

# Non-Abelian symmetry-resolved entanglement entropy

Eugenio Bianchi[1,2]*, Pietro Dona[3]† and Rishabh Kumar[1,2]‡

**1** Department of Physics, The Pennsylvania State University,
University Park, Pennsylvania 16802, USA
**2** Institute for Gravitation and the Cosmos, The Pennsylvania State University,
Pennsylvania 16802, USA
**3** Aix-Marseille Univ, Université de Toulon, CNRS, CPT, Marseille, France

* ebianchi@psu.edu , † pietro.dona@cpt.univ-mrs.fr , ‡ rishabh.kumar@psu.edu

## Abstract

We introduce a mathematical framework for symmetry-resolved entanglement entropy with a non-Abelian symmetry group. To obtain a reduced density matrix that is block-diagonal in the non-Abelian charges, we define subsystems operationally in terms of subalgebras of invariant observables. We derive exact formulas for the average and the variance of the typical entanglement entropy for the ensemble of random pure states with fixed non-Abelian charges. We focus on compact, semisimple Lie groups. We show that, compared to the Abelian case, new phenomena arise from the interplay of locality and non-Abelian symmetry, such as the asymmetry of the entanglement entropy under subsystem exchange, which we show in detail by computing the Page curve of a many-body system with $SU(2)$ symmetry.



# 1   Introduction

Symmetries play a fundamental role in isolated quantum systems as they result in conservation laws and constraints for physical quantities, including the entanglement entropy. In this paper, we study the interplay between locality, symmetries and entanglement. In particular, we show that the Page curve for the typical entanglement entropy [1,2] captures new phenomena proper of systems with a non-Abelian symmetry group [3].

For Abelian symmetries, such as number conservation or charge conservation, the notion of typical entanglement entropy [4–12] and its relation to symmetry-resolved entanglement [13–15] is well studied. The main ingredients are immediate to define. For instance, consider a system composed of two parts $A$ and $B$, which carry a representation of an Abelian group with charge $Q = Q_A + Q_B$. A symmetry-resolved state is an eigenstate of the total charge, $Q|\psi_q\rangle = q|\psi_q\rangle$, and the Hilbert space at fixed total charge $q$ decomposes as a direct sum of tensor products

$$\mathcal{H}^{(q)} = \bigoplus_{q_A} \left( \mathcal{H}_A^{(q_A)} \otimes \mathcal{H}_B^{(q-q_A)} \right). \tag{1}$$

The direct sum over subsystem charges $q_A$ is a consequence of the constraint $q = q_A + q_B$ imposed by charge conservation. To evaluate the entanglement entropy $S_A = -\mathrm{tr}_A(\rho_A \log \rho_A)$ of the pure state $|\psi_q\rangle$, we first compute the density matrix $\rho_A$ of the restricted state, which is defined by the partial trace over $B$ as usual [16], $\rho_A = \mathrm{tr}_B |\psi_q\rangle\langle\psi_q|$. Note that, as $\left[|\psi_q\rangle\langle\psi_q|, Q\right] = 0$, we have that the reduced density matrix commutes with the charge in $A$, i.e., $[\rho_A, Q_A] = 0$, and therefore it takes the block-diagonal form $\rho_A = \bigoplus_{q_A} p^{(q_A)} \rho_A^{(q_A)}$ with each block of definite charge $q_A$ having probability $p^{(q_A)}$. The generalization to a non-Abelian symmetry group is not immediate and requires new tools, which we introduce in this paper.

To illustrate the new aspects that arise in the presence of a non-Abelian symmetry, consider, for instance, a composite system that is invariant under the non-Abelian symmetry group $SU(2)$, with generators $\vec{J} = \vec{J}_A + \vec{J}_B$. How do we define a symmetry-resolved state? Clearly, it cannot be a simultaneous eigenstate of the components $J^x, J^y, J^z$ as these observables do

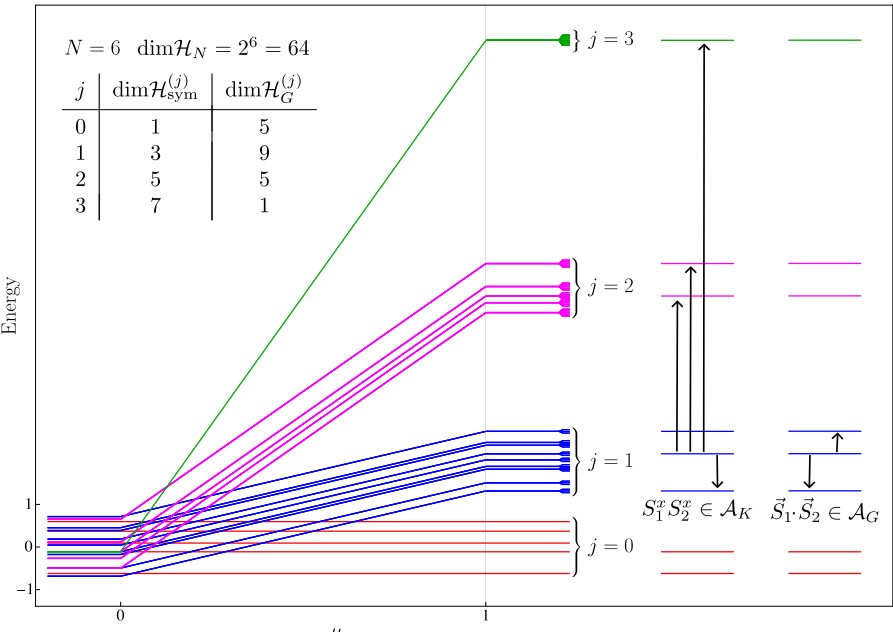

Figure 1: Energy spectrum of the $SU(2)$-invariant random Heisenberg Hamiltonian (2) with $N = 6$. On increasing the parameter $\mu$, the spectrum splits in symmetry-resolved blocks $\mathcal{H}_N^{(j)}$ of fixed spin $j$, (4), with each block consisting of $(2j+1) \times \dim \mathcal{H}_G^{(j)}$ states as shown in the table. We show also the transitions induced by the local observable $S_1^x S_2^x$ and by the $G$-invariant local observable $\vec{S}_1 \cdot \vec{S}_2$.

not commute. The only simultaneous eigenstates have $\vec{J}^2 = 0$. Even if we were to diagonalize only a set of commuting observables such as $\vec{J}^2$ and $J^z$, we still face the issue that the non-Abelian charges are not additive over subsystems as, for instance, $\vec{J}^2 \neq \vec{J}_A^2 + \vec{J}_B^2$. In the Abelian case, the proof that $\rho_A$ is block diagonal uses the additivity of the charges $Q = Q_A + Q_B$. Is $\rho_A$ block diagonal in the non-Abelian case? This question is related to how we define a subsystem. Do we measure only the group-invariant degrees of freedom or also the rotational degrees of freedom? We address these questions in detail, developing a mathematical framework for symmetry-resolved entanglement that applies to a general non-Abelian symmetry group $G$. We introduce the notions of (i) symmetry-resolved states and observables, (ii) locality and symmetry-resolved subsystems, and (iii) symmetry-resolved entanglement entropy and its statistics over random symmetry-resolved states.

As a concrete example of the interplay of locality, non-Abelian symmetry, and entanglement, let us consider a system of $N$ spin-1/2 particles with $SU(2)$ invariant Hamiltonian. Our mathematical analysis only requires the symmetry group $G$, and the Hamiltonian is used here simply to provide a physical motivation. For instance, we can consider the random Heisenberg Hamiltonian [17, 18]

$$H = \frac{1}{\sqrt{N}} \sum_{\substack{n,m=1 \\ n<m}}^{N} c_{nm} \vec{S}_n \cdot \vec{S}_m + \mu \vec{J}^2, \qquad \text{with} \quad \vec{J} = \sum_{n=1}^{N} \vec{S}_n, \tag{2}$$

where the coupling constants $c_{nm}$ are assumed to be normally distributed with zero average and unit variance, $\vec{S}_n$ are the individual spin operators, and $\mu$ is a coupling constant. The system is invariant under global $SU(2)$ rotations generated by $\vec{J}$,

$$[H, J^i] = 0. \tag{3}$$

This example allows us to illustrate the three notions that play a central role in this paper:

(i) *Symmetry-resolved states and observables*. Due to the presence of symmetry, the spectrum of the Hamiltonian splits into symmetry-resolved sectors corresponding to the eigenvalues of the conserved charge $\vec{J}^2$. Mathematically, the Hilbert space $\mathcal{H}_N$ of the system decomposes as a direct sum over sectors of spins $j$,

$$\mathcal{H}_N = \bigoplus_j \mathcal{H}_N^{(j)} = \bigoplus_j \big(\mathcal{H}_{\text{sym}}^{(j)} \otimes \mathcal{H}_G^{(j)}\big). \tag{4}$$

As shown in Fig. 1, each sector $\mathcal{H}_N^{(j)}$ of spin $j$ consists of $(2j+1) \times \dim \mathcal{H}_G^{(j)}$ orthogonal states, where the $(2j+1)$-multiplets describe the rotational degrees of freedom $|j,m\rangle \in \mathcal{H}_{\text{sym}}^{(j)}$ of the system. The internal degrees of freedom are rotational invariant states $|j,\chi_G\rangle \in \mathcal{H}_G^{(j)}$. A symmetry-resolved state $|\psi_j\rangle$ is defined as a state of the factorized form,

$$|\psi_j\rangle = |j,\xi\rangle |j,\chi_G\rangle, \qquad \text{with} \qquad |j,\xi\rangle = \sum_m \xi_m |j,m\rangle. \tag{5}$$

States of this form have been considered also in [19–21]. We define symmetry-resolved observables $O_G$, i.e., rotationally invariant observables that commute with the symmetry generators, $[O_G, J^i] = 0$. Note that the symmetry-resolved states defined by (5) are not generic states in the sector of spin $j$, as they have no entanglement between the rotational state $|j,m\rangle$ and the internal state $|j,\chi_G\rangle$. This definition is justified by the notion of symmetry-resolved observables $O_G$, which cannot entangle rotational and internal degrees of freedom. Moreover, if the state can only be prepared and measured with the observables $O_G$, then the rotational states $|j,m\rangle$ serve only as an ancilla, and the states of interest are the ones described above. In the following, we denote by $\mathcal{A}_K$ the algebra of observables of the system and by $\mathcal{A}_G \subset \mathcal{A}_K$ the subalgebra of $G$-invariant observables. While $\mathcal{A}_K$ is defined at the kinematical level, the algebra $\mathcal{A}_G$ knows about the symmetries of the dynamics.

To illustrate these definitions, let us suppose a large energy gap exists between the different sectors of spin $j$ because the parameter $\mu$ in the Hamiltonian is large. Then, the observables $O_G$ can induce only low-energy transitions because they have vanishing matrix elements between different sectors in the energy spectrum. Furthermore, they satisfy the selection rule $\Delta m = 0$ because they are rotationally invariant. States prepared and measured with these accessible observables $O_G$ take the form (5).

(ii) *Locality and symmetry-resolved subsystems*. A spin system like the one described by the Hamiltonian (2) has a built-in notion of locality associated with the single spins $\vec{S}_n$ that compose the system. These local subsystems come with a notion of tensor product decomposition $\mathcal{H} = \mathcal{H}_A \otimes \mathcal{H}_B$ for spins in the subsystems $A$ and $B$. The observables $O_A$ that act only on the subsystem $A$ form the subalgebra $\mathcal{A}_{KA} \subset \mathcal{A}_K$ of $K$-local observables in $A$: They are kinematically local, but they do not preserve the symmetries of the dynamics. To define symmetry-resolved subsystems that are local in $A$, we consider observables $O_{GA}$ that are both $G$-invariant and act only on the degrees of freedom in $A$. They form the subalgebra of $G$-local observables in $A$, given by the intersection:

$$\mathcal{A}_{GA} = \mathcal{A}_{KA} \cap \mathcal{A}_G. \tag{6}$$

As an example, the observable $S_1^x S_2^x \in \mathcal{A}_{KA}$ and the observable $\vec{S}_1 \cdot \vec{S}_2 \in \mathcal{A}_{GA}$ act locally on the first two spins only, but the first induces transitions between multiplets in the energy spectrum while the second does not as it is $G$-invariant.

A subsystem can be defined operationally in terms of a subalgebra of observables we can access. Here, we are interested in $G$-local observables in $A$. The restriction of a symmetry-resolved state $|\psi_j\rangle$ to the subalgebra $\mathcal{A}_{GA}$ is given by the density matrix $\rho_{GA}$. As $G$-local observables in $A$ commute with the generator of rotations in $A$, we have that $[\rho_{GA}, J_A^i] = 0$ and

therefore the reduced density matrix takes the block-diagonal form

$$\rho_{GA} = \bigoplus_{j_A} p^{(j_A)} \rho_{GA}^{(j_A)}. \tag{7}$$

Note that the choice of a $G$-local subalgebra is crucial here as it is associated with a decomposition of the Hilbert space $\mathcal{H}_N^{(j)}$ as a direct sum over $j_A$,

$$\mathcal{H}_N^{(j)} = \mathcal{H}_{\text{sym}}^{(j)} \otimes \bigoplus_{j_A} \left( \mathcal{H}_{GA}^{(j_A)} \otimes \mathcal{H}_{GB}^{(j,j_A)} \right). \tag{8}$$

Note also that this structure differs from the one found in the Abelian case: the Hilbert space of rotational degrees of freedom $\mathcal{H}_{\text{sym}}^{(j)}$ is non-trivial in the non-Abelian case, and the complement $\mathcal{H}_{GB}^{(j,j_A)}$ is not labeled by a difference of the charges $j$ and $j_A$ as in $\mathcal{H}_B^{(q-q_A)}$ in Eq. (1). Furthermore, if instead we had considered the usual reduced density matrix $\rho_{KA} = \text{tr}_B |\psi_j\rangle\langle\psi_j|$, a different structure would arise. The kinematical density matrix $\rho_{KA}$ is defined as a trace over the kinematical degrees of freedom in $B$ and represents the restriction of the state to the subalgebra of $K$-local observables $\mathcal{A}_{KA}$. We note that in the non-Abelian case the $K$-local reduced density matrix $\rho_{KA}$ is not of a block-diagonal form, as $\vec{J}^2 \neq \vec{J}_A^2 + \vec{J}_B^2$ and $K$-local observables induce transitions between sectors with different spins $j_A$ in the subsystem $A$. In Fig. 2, we also show that considering instead a microcanonical truncation of local observables [22] by introducing projectors $P^{(j)}$, we can forbid transitions between sectors at the expense of locality. The operational definition of a subsystem in terms of the subalgebra of $G$-local observables guarantees that the subsystem is both local and $G$-invariant.

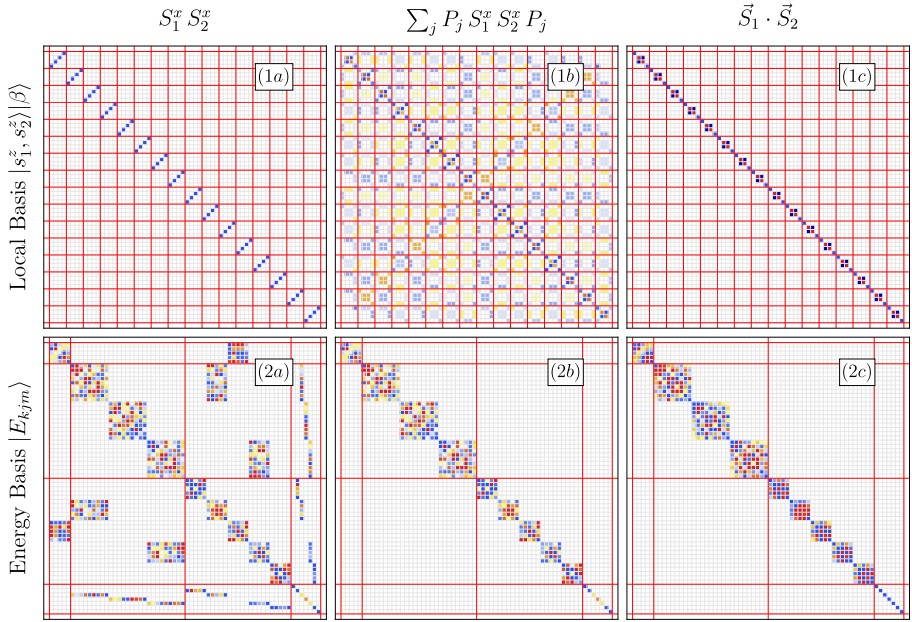

Figure 2: Matrix elements of the $K$-local observable $S_1^x S_2^x$ (1a)–(2a), of its $SU(2)$-projected version $\sum_j P_j S_1^x S_2^x P_j$ (1b)–(2b), and of the $G$-local observable $\vec{S}_1 \cdot \vec{S}_2$ (1c)–(2c). Block-diagonal matrix elements in the spin-lattice basis (1a) highlight the local or non-local nature of the observable, while the off-diagonal matrix elements in the energy basis (2a) determine the possible transitions between energy levels. While the $SU(2)$-projected observable allows only $SU(2)$-invariant transitions ($\in \mathcal{A}_G$) (2b), it is not a local observable ($\notin \mathcal{A}_{KA}$) (1b). Hence, the $SU(2)$-projected observable is not $G$-local, (6). On the other hand, the observable $\vec{S}_1 \cdot \vec{S}_2 \in \mathcal{A}_{GA}$ is both local (1c) and induces only $SU(2)$-invariant transitions (2c).

(iii) *Symmetry-resolved entanglement entropy and typicality*. If we have access only to symmetry-resolved observables in the subsystem $A$, i.e., to the subalgebra $\mathcal{A}_{GA}$, then the accessible entanglement entropy is defined by the restriction of the state to this subalgebra. For a symmetry-resolved state $|\psi_j\rangle$, the accessible entropy is the symmetry-resolved entanglement entropy $S_{GA}$ defined as the von Neumann entropy of the density matrix $\rho_{GA}$, which in turn can be expressed as the sum of two terms:

$$S_{GA}(|\psi_j\rangle) = -\text{tr}(\rho_{GA}\log\rho_{GA}) = -\sum_{j_A} p^{(j_A)}\text{tr}(\rho_{GA}^{(j_A)}\log\rho_{GA}^{(j_A)}) - \sum_{j_A} p^{(j_A)}\log p^{(j_A)}. \quad (9)$$

The symmetry-resolved entanglement entropy $S_{GA}(|\psi_j\rangle)$ can be understood as the sum of two terms, $S_{GA}^{(\text{conf})}$ and $S_{GA}^{(\text{num})}$. One first defines the symmetry-resolved entanglement entropy at fixed system *and* subsystem charges $S_{GA}^{(j_A)}$ as the von Neumann entropy of the density matrix $\rho_{GA}^{(j_A)}$ in each sector $j_A$. Then the configurational entropy $S_{GA}^{(\text{conf})}$ is given by its average over the probability $p^{(j_A)}$ of finding the state in the sector $j_A$, and the number entropy $S_{GA}^{(\text{num})}$ is the Shannon entropy of the probability $p^{(j_A)}$. The table below summarizes the definitions commonly used in the literature:

| | |
|---|---|
| Total symmetry-resolved entanglement entropy | $S_{GA}(|\psi_j\rangle)$ |
| Symmetry-resolved entanglement entropy at fixed system & subsystem charges | $S_{GA}^{(j_A)} = -\text{tr}(\rho_{GA}^{(j_A)}\log\rho_{GA}^{(j_A)})$ |
| Configurational entropy | $S_{GA}^{(\text{conf})} = \sum_{j_A} p^{(j_A)}S_{GA}^{(j_A)}$ |
| Number entropy (Shannon) | $S_{GA}^{(\text{num})} = -\sum_{j_A} p^{(j_A)}\log p^{(j_A)}$ |

$$(10)$$

Now, given a random state $|\psi_j\rangle$ that is symmetry resolved, we can ask what is the probability of finding entanglement entropy $S_{GA}$. It turns out that the exact formulas for the average $\langle S_{GA}\rangle$ and the variance $(\Delta S_{GA})^2$ derived for a general subalgebra of observables in [2] apply here unmodified, once one uses the dimensions of the Hilbert spaces appearing in the decomposition (8). For a system composed of $N \gg 1$ spins, there is a typical value $\langle S_{GA}\rangle$ of the entropy that characterizes completely the Page curve of subsystems [1, 2] as the variance of the probability distribution $P(S_{GA})$ is exponentially small in $N$.

It is important to note that the $G$-invariant notion of entanglement entropy $S_{GA}$ introduced here is distinct from the usual notion of kinematical entanglement entropy $S_{KA} = -\text{tr}(\rho_{KA}\log\rho_{KA})$ which probes also the rotational degrees of freedom $|j, m\rangle$ that are not $G$-invariant. In Fig. 3, we illustrate the difference with an example. We consider the case of $N = 6$ spin-1/2 particles in a random state with $j = 1$ and $m = +1$, and a subsystem consisting of $N_A = 3$ spins. For the same sample of random states, we report the histograms for the probability distributions $P(S_{GA})$ and $P(S_{KA})$, which are shown to be distinct, Fig. 3(a). The statistical average $\mu_{GA}$ and variance $\sigma_{GA}^2$ match numerically the exact formulas $\langle S_{GA}\rangle$ and $(\Delta S_{GA})^2$ that we derive in this paper for the $G$-local entanglement entropy. The $K$-local entanglement entropy with $SU(2)$ symmetry is studied in [21] where asymptotic formulas at large $N$ are derived for $\langle S_{KA}\rangle$ and $(\Delta S_{KA})^2$ using a combination of analytical and numerical methods. To date, there is no analytical result for the average $K$-local entanglement entropy $\langle S_{KA}\rangle$ at finite $N$. We note that, because of the contribution of the rotational degrees of freedom $|j, m\rangle$, we have that typically, the kinematical entanglement entropy $S_{KA}$ is larger than the symmetry-resolved entanglement entropy $S_{GA}$ of a random state. However, the relation between the two is non-trivial as shown in Fig. 3(b) where we consider a specific (non-random) family of states for which $S_{GA}$ is instead larger than $S_{KA}$. We consider the two orthogonal states $|\psi_1\rangle$ and $|\psi_2\rangle$

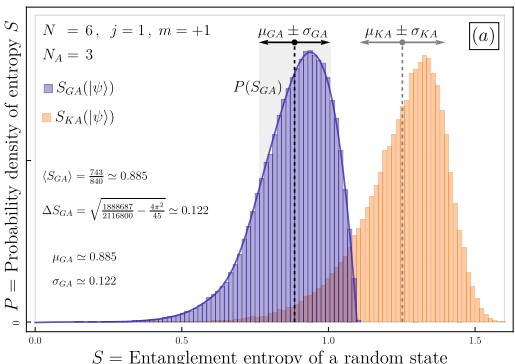
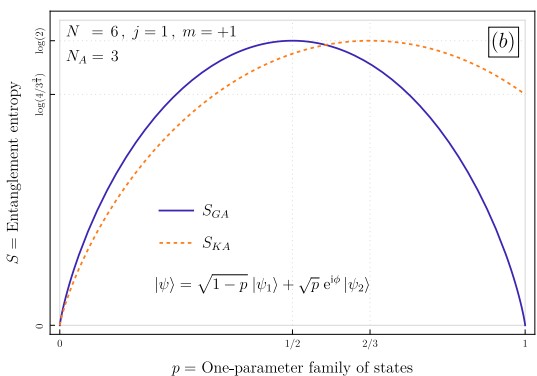

Figure 3: For a system of $N = 6$ spins with total spin $j = 1$, magnetization $m = +1$ and subsystem size $N_A = 3$, we show: $(a)$ the probability distribution of the $G$-local (purple) and the $K$-local (yellow) entanglement entropies $S_{GA}$ and $S_{KA}$ of a sample of random symmetry-resolved states, including a comparison of the numerical values $(\mu_{GA}, \sigma_{GA})$ and the exact values $(\langle S_{GA}\rangle, \Delta S_{GA})$ of the average and variance of $P(S_{GA})$; $(b)$ the entanglement entropy of a superposition of the two states (11), which highlights the non-trivial relation between $S_{GA}$ and $S_{KA}$; at $p = 0$: $S_{GA} = S_{KA} = 0$, at $p = \frac{1}{2}$: $S_{GA} > S_{KA}$, and at $p = 1$: $S_{KA} > S_{GA} = 0$.

with $j = 1$, $m = +1$ obtained from the coupling of the angular momentum of six spin-$1/2$ particles as described by the diagrams below [23]:

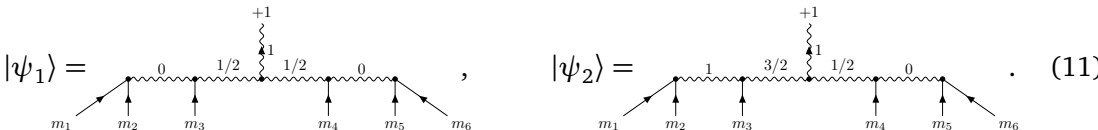

$$|\psi_1\rangle = \quad\quad\quad , \quad\quad |\psi_2\rangle = \quad\quad\quad . \quad (11)$$

These two states are simultaneous eigenstates of the observables $(\vec{S}_1 + \vec{S}_2)^2$ and $(\vec{S}_1 + \vec{S}_2 + \vec{S}_3)^2$. As a result, when restricted to $G$-invariant observables of the first three spins, they have vanishing $G$-local entropy $S_{GA}(|\psi_1\rangle) = 0$ and $S_{GA}(|\psi_2\rangle) = 0$. On the other hand, because of the entanglement in the rotational degrees of freedom, we have that the $K$-local entropies are $S_{KA}(|\psi_1\rangle) = 0$ and non-vanishing $S_{KA}(|\psi_2\rangle) \neq 0$. Moreover, in the superposition $(|\psi_1\rangle + |\psi_2\rangle)/\sqrt{2}$ we have that $S_{GA}$ is larger than $S_{KA}$, which shows that the relation between the two is non-trivial.

The three ingredients (i)–(ii)–(iii) outlined above are related to a substantial body of literature. The notion of Page curve for random states without constraints was first introduced in [1] motivated by the information puzzle in black hole evaporation [24–26]. In the case of Abelian constraints such as number conservation [2], the notions of $K$-local and $G$-local subsystems coincide, and the Page curve with Abelian constraints is studied in [4–12]. In the case of non-Abelian symmetry $SU(2)$, the Page curve of $K$-local subsystems is studied in [21, 27]. The operational notion of entanglement entropy associated with a subalgebra of observables is discussed in [28–31]. In particular, the geometric entanglement entropy in quantum field theory [32–34] is best understood in terms of local subalgebras of observables [35–39]. The notion of $G$-local subalgebras of observables invariant under a group of non-Abelian transformations appears in lattice gauge theory [40–43] and in loop quantum gravity [44–46]. The accessible entanglement entropy in the presence of symmetries and superselections rules is discussed in [47–53]. The notion of symmetry-resolved entanglement was introduced in [13–15] and studied for Abelian symmetry in [54–70] and [71–83], (see [84] for a review). To date, the study of symmetry-resolved entanglement entropy for non-Abelian groups has remained restricted to vacuum states that are assumed to be invariant under the action of the group, such as

the $j = 0$ sector in the decomposition (8) for the group $SU(2)$ [14, 85–87], or the $|0,0\rangle$ vacuum that is invariant under Virasoro symmetry in a conformal field theory [88]. On the other hand, the case of excited states with symmetry group $SU(2)$ has been studied recently in [19–21] where non-commuting conserved charges are shown to lead to new phenomena [27, 89–91] in the context of eigenstate thermalization [92–94] and quantum many-body scars [95, 96].

The paper is organized as follows: In Sec. 2 we describe how to define a subsystem operationally in terms of a subalgebra of observables, and we determine the main expression for the typical entanglement entropy in complete generality. Then, in Sec. 3, we introduce a non-Abelian symmetry group, and we derive the decomposition (7)–(8) for general group $G$, without any requirement of locality. In Sec. 4, we discuss locality and introduce the notion of $G$-local observables in many-body systems using the definition (6). The application to many-body spin systems with $SU(2)$ symmetry is discussed in Sec. 5, where we derive the exact formulas used in Fig. 3 (see also App. A). In Sec. 6, we derive the large system asymptotics of the $SU(2)$ symmetry-resolved typical entropy (see also App. B). To conclude, in Sec. 7, we summarize our results and illustrate applications.

## 2 Subsystems from subalgebras and typical entropy

We consider an isolated quantum system with Hilbert space $\mathcal{H}$ of finite dimension $D = \dim \mathcal{H} < \infty$. Pure states of the system, $|\psi\rangle \in \mathcal{H}$, can be understood as vectors in $\mathbb{C}^D$ and observables $O$ as $D \times D$ hermitian matrices. The algebra of observables of the system,

$$\mathcal{A} = \mathcal{L}(\mathcal{H}) \simeq M_D(\mathbb{C}), \tag{12}$$

is the set $\mathcal{L}(\mathcal{H})$ of linear operators on $\mathcal{H}$ or equivalently the algebra $M_D(\mathbb{C})$ of matrices on $\mathbb{C}^D$.

In general, a von Neumann algebra on $\mathbb{C}^D$ is an algebra of matrices that is closed under (i) hermitian conjugation, (ii) addition, (iii) multiplication, and (iv) contains all $\mathbb{C}$ multiples of the identity operator. In particular, the algebra $\mathcal{A}$ of observables of the system is a von Neumann algebra [29, 97].[1] Here we are interested in von Neumann subalgebras of $\mathcal{A}$. We say that a subalgebra is generated by the set of hermitian matrices $K_i$ (with $i = 1, \ldots, n$) if it can be obtained by their closure under (i)–(iv), which we denote by

$$\mathbb{C}[K_i] \equiv \left\{ M \in M_D(\mathbb{C}) \,\middle|\, M = a\mathbb{1} + \sum_i b_i K_i + \sum_{ij} c_{ij} K_i K_j + \ldots \right\}, \tag{13}$$

with coefficients $a, b_i, c_{ij} \ldots$ in $\mathbb{C}$. The notion of commutant will play a central role. The commutant of a set of matrices is defined as

$$\{K_1, \ldots, K_n\}' \equiv \{M \in M_D(\mathbb{C}) \,|\, MK_i - K_i M = 0, \ i = 1, \ldots, n\}. \tag{14}$$

There are two useful results that we will use multiple times: the first relates the the double commutant of a set of matrices to the algebra they generate, $\{K_1, \ldots, K_n\}'' = \mathbb{C}[K_1, \ldots, K_n]$; the second result relates the intersection of commutants to the union of the sets of generators:

$$\mathbb{C}[K_i]' \cap \mathbb{C}[H_j]' = \mathbb{C}[K_i, H_j]'. \tag{15}$$

Note that the commutant of a von Neumann algebra $\mathcal{A}_S \subset \mathcal{A}$ is also a von Neumann algebra $(\mathcal{A}_S)' \subset \mathcal{A}$, and the double commutant coincides with the algebra itself, $(\mathcal{A}_S)'' = \mathcal{A}_S$.

---

[1]While observables of the system are represented by hermitian matrices, linear combinations are assumed to be over $\mathbb{C}$ and therefore the von Neumann algebra of observables contains also anti-hermitian matrices and, more importantly, unitary transformations. We note also that here we have assumed that the Hilbert space of the system is finite-dimensional, and therefore in the definition of a von Neumann algebra reduces to the one of a matrix algebra [97], with no additional requirement about topological closure. For a discussion of the infinite-dimensional case and the classification of von Neumann algebras of observables in quantum field theory, we refer to [35–37, 98].

## 2.1 Hilbert space decomposition adapted to a subalgebra of observables

Given our physical system, we define a *subsystem* $\mathcal{S}$ operationally, in terms of a set of observables $\{O_1, O_2, \dots\}$ that we have access to or that we can probe with measuring devices available to us. These observables generate a von Neumann algebra $\mathcal{A}_{\mathcal{S}}$ on $\mathcal{H}$,

$$\mathcal{A}_{\mathcal{S}} = \mathbb{C}[O_1, O_2, \dots], \tag{16}$$

that is a subalgebra of the algebra of observables $\mathcal{A}$ of the system, $\mathcal{A}_{\mathcal{S}} \subseteq \mathcal{A}$. The two notions of commutant and center allow us to decompose the Hilbert space $\mathcal{H}$ in sectors adapted to the subalgebra of observables $\mathcal{A}_{\mathcal{S}}$.[2]

We start with the notion of commutant. We denote by $\overline{\mathcal{S}}$ the *complement* of the subsystem $\mathcal{S}$ and define it in terms of the subalgebra of observables that commute with $\mathcal{A}_{\mathcal{S}}$, i.e., the commutant of $\mathcal{A}_{\mathcal{S}}$ in $\mathcal{A}$,

$$\mathcal{A}_{\overline{\mathcal{S}}} = (\mathcal{A}_{\mathcal{S}})' \equiv \{M \in \mathcal{A} \mid [M, N] = 0, \ \forall N \in \mathcal{A}_{\mathcal{S}}\}. \tag{17}$$

Next, we consider the *center* $\mathcal{Z}_{\mathcal{S}}$ of the subalgebra. Note that in general there are observables $R_1, R_2, \dots$ that can be measured both from $\mathcal{S}$ and from $\overline{\mathcal{S}}$, i.e. the subalgebra $\mathcal{A}_{\mathcal{S}}$ can have a non-trivial center $\mathcal{Z}_{\mathcal{S}}$,

$$\mathcal{Z}_{\mathcal{S}} = \mathcal{A}_{\mathcal{S}} \cap \mathcal{A}_{\overline{\mathcal{S}}} = \mathbb{C}[R_1, R_2, \dots]. \tag{18}$$

These structures allow us to decompose the Hilbert space $\mathcal{H}$ as a direct sum of tensor products. The construction can be understood concretely in terms of an orthonormal basis of $\mathcal{H}$ adapted to the subsystem $\mathcal{S}$. We first consider the observables $R_1, R_2, \dots$ in the center $\mathcal{Z}_{\mathcal{S}}$. By definition, the center $\mathcal{Z}_{\mathcal{S}}$ is an Abelian algebra and, therefore, we can diagonalize these observables simultaneously. We denote by $r$ the eigenvalues of the observables in the center $\mathcal{Z}_{\mathcal{S}}$. Then, we can select a maximal commuting set of observables in $\mathcal{A}_{\mathcal{S}}$ with simultaneous eigenvalues $\alpha$, and a maximal commuting set of observables in $\mathcal{A}_{\overline{\mathcal{S}}}$ with simultaneous eigenvalues $\beta$. As a result, we obtain an orthonormal basis of $\mathcal{H}$,

$$|r, \alpha, \beta\rangle = |r, \alpha\rangle |r, \beta\rangle, \tag{19}$$

that is adapted to the subalgebra $\mathcal{A}_{\mathcal{S}}$. This basis gives a concrete meaning to the direct sum decomposition

$$\mathcal{H} = \bigoplus_r \left( \mathcal{H}_{\mathcal{S}}^{(r)} \otimes \mathcal{H}_{\overline{\mathcal{S}}}^{(r)} \right), \tag{20}$$

where $|r, \alpha\rangle$ is an orthonormal basis of $\mathcal{H}_{\mathcal{S}}^{(r)}$, and $|r, \beta\rangle$ an orthonormal basis of $\mathcal{H}_{\overline{\mathcal{S}}}^{(r)}$. Note that when the center is trivial, i.e., $\mathcal{Z}_{\mathcal{S}} = \{\mathbb{1}\}$, this decomposition reduces to the familiar tensor product structure $\mathcal{H} = \mathcal{H}_A \otimes \mathcal{H}_B$ of a composite system.

We call $d_r$ and $b_r$ the *dimensions* of the sectors appearing in the direct-sum decomposition associated with a subalgebra $\mathcal{A}_{\mathcal{S}}$,

$$d_r = \dim \mathcal{H}_{\mathcal{S}}^{(r)}, \qquad b_r = \dim \mathcal{H}_{\overline{\mathcal{S}}}^{(r)}, \qquad D = \dim \mathcal{H} = \sum_r d_r b_r. \tag{21}$$

These dimensions play a central role in the expression of the typical entropy of a subsystem.

---

[2]We refer to Ch. 11.8 in [29] for a pedagogical introduction, to [41–43] for applications to lattice systems, to [99] for applications to information transport, and to [100, 101] for applications to Hilbert space fragmentation.

## 2.2 Pure states restricted to a subalgebra and entanglement entropy

Using the decomposition (20), observables of the subsystem $\mathcal{S}$ take the direct-sum form

$$O \in \mathcal{A}_{\mathcal{S}} \subseteq \mathcal{A} \quad \Longrightarrow \quad O = \bigoplus_r \left( O_{\mathcal{S}}^{(r)} \otimes \mathbb{1}_{\bar{\mathcal{S}}}^{(r)} \right). \tag{22}$$

It is useful to introduce a notion of Hilbert space $\mathcal{H}_{\mathcal{S}}$ of the subsystem, defined as the direct sum of the subsystem sectors:

$$\mathcal{H}_{\mathcal{S}} \equiv \bigoplus_r \mathcal{H}_{\mathcal{S}}^{(r)}. \tag{23}$$

The restriction of an operator $O \in \mathcal{A}_{\mathcal{S}}$ to the Hilbert space of the subsystem is given by

$$O \in \mathcal{A}_{\mathcal{S}} \subseteq \mathcal{A} \quad \Longrightarrow \quad O_{\mathcal{S}} \equiv \bigoplus_r O_{\mathcal{S}}^{(r)} \ \in \ \mathcal{L}(\mathcal{H}_{\mathcal{S}}). \tag{24}$$

We can write this map from $O$ to $O_{\mathcal{S}}$ concretely as

$$\mathrm{Tr}(\,\cdot\,\Pi_{\mathcal{S}}): \quad \mathcal{L}(\mathcal{H}) \longrightarrow \mathcal{L}(\mathcal{H}_{\mathcal{S}}), \tag{25}$$

$$O \longmapsto O_{\mathcal{S}} = \mathrm{Tr}(O\,\Pi_{\mathcal{S}}),$$

where Tr is the trace over $\mathcal{H}$ and $\mathrm{Tr}_{\mathcal{S}}$ is the trace over $\mathcal{H}_{\mathcal{S}}$.[3] In the adapted basis $|r,\alpha,\beta\rangle \in \mathcal{H}$ and $|r,\alpha\rangle \in \mathcal{H}_{\mathcal{S}}$, the map $\Pi_{\mathcal{S}}$ takes the form

$$\Pi_{\mathcal{S}} = \sum_r \sum_{\alpha\alpha'} \left( \sum_\beta |r,\alpha,\beta\rangle\langle r,\alpha',\beta| \right) \otimes |r,\alpha'\rangle\langle r,\alpha|. \tag{26}$$

With these definitions, we can now write the restriction of a pure state $|\psi\rangle \in \mathcal{H}$ to the subsystem $\mathcal{S}$ as a density matrix $\rho_{\mathcal{S}}$,

$$\rho = |\psi\rangle\langle\psi| \quad \Longrightarrow \quad \rho_{\mathcal{S}} = \mathrm{Tr}(\rho\,\Pi_{\mathcal{S}}), \tag{27}$$

so that the expectation value of an observable on the subsystem $\mathcal{S}$ is given by

$$O \in \mathcal{A}_{\mathcal{S}} \subseteq \mathcal{A} \quad \Longrightarrow \quad \langle\psi|O|\psi\rangle = \mathrm{Tr}_{\mathcal{S}}(O_{\mathcal{S}}\,\rho_{\mathcal{S}}). \tag{28}$$

We note that the map $\rho \mapsto \rho_{\mathcal{S}}$ is completely positive and trace preserving (CPTP) [16], as can be seen by writing it in the operator sum form $\mathrm{Tr}(\rho\,\Pi_{\mathcal{S}}) = \sum_{r\beta} Y_{r\beta}^\dagger \rho\, Y_{r\beta}$ with Kraus operators $Y_{r\beta} = \sum_\alpha |r,\alpha,\beta\rangle\langle r,\alpha|$.

We are now ready to define the entanglement entropy $S$ of the pure state $|\psi\rangle \in \mathcal{H}$ restricted to a subalgebra of observables $\mathcal{A}_{\mathcal{S}} \subseteq \mathcal{A} = \mathcal{L}(\mathcal{H})$: the entropy of the triple $\left(|\psi\rangle, \mathcal{A}, \mathcal{A}_{\mathcal{S}}\right)$ equals the von Neumann entropy $S_{\mathrm{vN}}(\rho_{\mathcal{S}})$ of the restricted state, i.e.,

$$S\left(|\psi\rangle, \mathcal{A}, \mathcal{A}_{\mathcal{S}}\right) = -\mathrm{Tr}_{\mathcal{S}}(\rho_{\mathcal{S}} \log \rho_{\mathcal{S}}), \qquad \text{with} \qquad \rho_{\mathcal{S}} = \langle\psi|\Pi_{\mathcal{S}}|\psi\rangle. \tag{29}$$

---

[3]Traces over the Hilbert spaces $\mathcal{H}$, $\mathcal{H}_{\mathcal{S}}$ and $\mathcal{H}_{\mathcal{S}}^{(r)}$ are defined as

$$\mathrm{Tr}(\cdot) = \sum_{r\alpha\beta} \langle r,\alpha,\beta| \cdot |r,\alpha,\beta\rangle, \qquad \mathrm{Tr}_{\mathcal{S}}(\cdot) = \sum_r \sum_\alpha \langle r,\alpha| \cdot |r,\alpha\rangle, \qquad \mathrm{Tr}_{\mathcal{S}}^{(r)}(\cdot) = \sum_\alpha \langle r,\alpha| \cdot |r,\alpha\rangle.$$

Similarly, one can define traces for $\mathcal{H}_{\bar{\mathcal{S}}}^{(r)}$ and for $\mathcal{H}_{\bar{\mathcal{S}}} = \bigoplus_r \mathcal{H}_{\bar{\mathcal{S}}}^{(r)}$ in term of the basis $|r,\beta\rangle$.

It is useful to express the entanglement entropy in the basis (19) adapted to a subalgebra $\mathcal{A}_\mathcal{S}$. We introduce first the projector $P^{(r)} = \sum_{\alpha\beta} |r,\alpha,\beta\rangle\langle r,\alpha,\beta|$ to the sector $r$ of the Hilbert space decomposition. Using this projector, we can define the probability $p_r$ that the pure state is found to be in the sector $r$, together with the (normalized) projected state $|\psi^{(r)}\rangle$:

$$p_r = \langle\psi|P^{(r)}|\psi\rangle = \sum_{\alpha\beta} \left|\langle r,\alpha,\beta|\psi\rangle\right|^2, \tag{30}$$

$$|\psi^{(r)}\rangle = \frac{P^{(r)}|\psi\rangle}{\sqrt{\langle\psi|P^{(r)}|\psi\rangle}} = \sum_{\alpha\beta} \psi^{(r)}_{\alpha\beta} |r,\alpha\rangle|r,\beta\rangle. \tag{31}$$

Any pure state $|\psi\rangle \in \mathcal{H}$ can then be expressed in the basis adapted to the subalgebra of observables $\mathcal{A}_\mathcal{S}$ as

$$|\psi\rangle = \sum_r \sqrt{p_r} \sum_{\alpha\beta} \psi^{(r)}_{\alpha\beta} |r,\alpha\rangle|r,\beta\rangle. \tag{32}$$

The definition (27) of reduced density matrix takes then the direct sum form

$$\rho_\mathcal{S} = \bigoplus_r p_r \rho^{(r)}_\mathcal{S}, \tag{33}$$

with $\rho^{(r)}_\mathcal{S}$ simply defined as the partial trace over $\mathcal{H}^{(r)}_{\overline{\mathcal{S}}}$, i.e.,

$$\rho^{(r)}_\mathcal{S} = \mathrm{Tr}^{(r)}_{\overline{\mathcal{S}}}\left(|\psi^{(r)}\rangle\langle\psi^{(r)}|\right) = \sum_{\alpha\alpha'}\left(\sum_\beta \psi^{(r)}_{\alpha'\beta}\psi^{(r)*}_{\alpha\beta}\right)|r,\alpha'\rangle\langle r,\alpha|. \tag{34}$$

Using the decomposition (33) and expanding $\mathrm{Tr}_\mathcal{S}(\rho_\mathcal{S}\log\rho_\mathcal{S}) = \sum_r \mathrm{Tr}^{(r)}_\mathcal{S}\left(p_r\rho^{(r)}_\mathcal{S}\log(p_r\rho^{(r)}_\mathcal{S})\right)$, we can write the entanglement entropy as the sum of two terms,

$$S\left(|\psi\rangle, \mathcal{A}, \mathcal{A}_\mathcal{S}\right) = -\sum_r p_r \mathrm{tr}\left(\rho^{(r)}_\mathcal{S}\log\rho^{(r)}_\mathcal{S}\right) - \sum_r p_r \log p_r, \tag{35}$$

where the first term is the average over sectors of the entanglement entropy in each sector, and the second piece is the Shannon entropy of the distribution over sectors.

We summarize some useful properties of the entanglement entropy of a subsystem defined in terms of a subalgebra:

**Minimum** — The entanglement entropy is non-negative and vanishes if and only if the restriction of the state to the subalgebra is pure. For this to happen, the state has to belong to a definite sector $r$ and have a product form, i.e.,

$$S\left(|\psi\rangle, \mathcal{A}, \mathcal{A}_\mathcal{S}\right) = 0 \quad\Longrightarrow\quad \exists\, r: \quad |\psi\rangle = |r,\xi\rangle_\mathcal{S}|r,\chi\rangle_{\overline{\mathcal{S}}}, \tag{36}$$

with $|r,\xi\rangle_\mathcal{S} = \sum_\alpha \xi_\alpha|r,\alpha\rangle$ and $|r,\chi\rangle_{\overline{\mathcal{S}}} = \sum_\beta \chi_\beta|r,\beta\rangle$. As a special case, if we don't restrict the state to any subalgebra, then the entanglement entropy of a pure state vanishes, $S\left(|\psi\rangle, \mathcal{A}, \mathcal{A}\right) = 0$.

**Maximum** — We can determine the maximum entanglement entropy by varying the Lagrangian $L = -\sum_r p_r \sum_i (\lambda_{ri}\log\lambda_{ri}) - \sum_r p_r \log p_r + \mu_0(1 - \sum_r p_r) + \sum_r \mu_r(1 - \sum_i \lambda_{ri})$, with probabilities $p_r$ and $\lambda_{ri}$, and Lagrange multipliers $\mu_0$ and $\mu_r$. At the stationary point, we obtain the maximum entropy

$$S_{\max} = \log\left(\sum_r \min(d_r, b_r)\right). \tag{37}$$

The maximally-entangled state can be written as

$$|I\rangle = \sum_r \sqrt{\frac{\min(d_r, b_r)}{\sum_{r'} \min(d_{r'}, b_{r'})}} \sum_{i=1}^{\min(d_r, b_r)} \frac{1}{\sqrt{\min(d_r, b_r)}} |r, i\rangle_S |r, i\rangle_{\overline{S}}. \tag{38}$$

**Commutant symmetry** — Note that the entanglement entropy of a subsystem is symmetric under the exchange of the subsystem with its commutant, i.e.,

$$S(|\psi\rangle, \mathcal{A}, \mathcal{A}_{\overline{S}}) = S(|\psi\rangle, \mathcal{A}, \mathcal{A}_S). \tag{39}$$

This notion of commutant symmetry generalizes the familiar subsystem symmetry $S_A = S_B$ of the entanglement entropy of pure states in a tensor product $\mathcal{H}_A \otimes \mathcal{H}_B$.

## 2.3 Typical entanglement entropy: Average and variance

Consider the ensemble of random pure states $|\psi\rangle \in \mathcal{H}$. Given a function of the state, $f(|\psi\rangle)$, we can define the average over the ensemble as

$$\langle f \rangle_{\mathcal{H}} = \int_{\mathcal{H}} d\mu(\psi) f(|\psi\rangle) = \int dU f(U|\psi_0\rangle), \tag{40}$$

where $d\mu(\psi)$ is the uniform measure over the unit sphere in $\mathbb{C}^D$ or, equivalently, $dU$ is the Haar measure over the unitary group $U(D)$ that allows us to write a random state $|\psi\rangle = U|\psi_0\rangle$ in terms of a reference state $|\psi_0\rangle$ and a random unitary $U$. The function $f$ can be a linear function of $|\psi\rangle\langle\psi|$, such as the expectation value of a subsystem observable in $\mathcal{A}_S$, or a non-linear function as the entanglement entropy $S(|\psi\rangle, \mathcal{A}, \mathcal{A}_S)$ of the state restricted to a subalgebra of observables. We note that the result of this average can be expressed purely in terms of the dimensions (21) of the sectors of the Hilbert space decomposition (20). For instance, the average density matrix of a pure state and of its restriction to a subsystem $S$ is

$$\rho = |\psi\rangle\langle\psi| \implies \langle \rho \rangle_{\mathcal{H}} = \frac{1}{D}\mathbb{1}, \qquad \langle \rho_S \rangle_{\mathcal{H}} = \mathrm{Tr}_S(\langle \rho \rangle_{\mathcal{H}} \Pi_S) = \bigoplus_r \frac{d_r b_r}{D} \frac{1}{d_r} \mathbb{1}_S^{(r)}, \tag{41}$$

which results in the von Neumann entropy of the average state

$$S_{\mathrm{vN}}(\langle \rho_S \rangle_{\mathcal{H}}) = -\mathrm{Tr}_S(\langle \rho_S \rangle_{\mathcal{H}} \log\langle \rho_S \rangle_{\mathcal{H}}) = \sum_r \frac{d_r b_r}{D} \log\left(\frac{D}{b_r}\right). \tag{42}$$

While in general, the average $\langle f \rangle_{\mathcal{H}}$ alone does not characterize the typical value of a function $f(|\psi\rangle)$ for a random state, computing its moments $\langle f^n \rangle_{\mathcal{H}}$ allows us to characterize the probability distribution. We are interested in the probability $P(S) dS$ that a random pure state $|\psi\rangle$ has entropy $S$ when restricted to the subalgebra $\mathcal{A}_S$. When the dispersion around the average is small, $\Delta S \ll \langle S \rangle$, we can simply use the average entropy $\langle S \rangle$ to characterize the entropy. In this case, we say that the *typical entanglement entropy* of a random pure state is $\langle S \rangle$.

In [2], the exact formulas for the average entanglement entropy $\langle S \rangle$ and its variance $(\Delta S)^2$ for a pure random state restricted to a subalgebra of observables corresponding to the Hilbert space decomposition (20) were found to be:

$$\langle S \rangle \equiv \langle S(|\psi\rangle, \mathcal{A}, \mathcal{A}_S)\rangle_{\mathcal{H}} = \sum_r \varrho_r \varphi_r, \tag{43}$$

$$(\Delta S)^2 \equiv \langle S^2 \rangle - \langle S \rangle^2 = \frac{1}{D+1}\left(\sum_r \varrho_r (\varphi_r^2 + \chi_r) - \left(\sum_r \varrho_r \varphi_r\right)^2\right), \tag{44}$$

with the quantities $\varrho_r$, $\varphi_r$, $\chi_r$ expressed in terms of the dimensions $d_r$, $b_r$, $D$ (21) given by

$$\varrho_r = \frac{d_r b_r}{D}, \tag{45}$$

$$\varphi_r = \begin{cases} \Psi(D+1) - \Psi(b_r+1) - \dfrac{d_r - 1}{2 b_r}, & d_r \leq b_r, \\[2ex] b_r \longleftrightarrow d_r, & d_r > b_r, \end{cases} \tag{46}$$

$$\chi_r = \begin{cases} (d_r + b_r)\Psi'(b_r+1) - (D+1)\Psi'(D+1) - \dfrac{(d_r-1)(d_r + 2b_r - 1)}{4 b_r^2}, & d_r \leq b_r, \\[2ex] b_r \longleftrightarrow d_r, & d_r > b_r, \end{cases} \tag{47}$$

where $\Gamma(x)$ is the gamma function, $\Psi(x) = \Gamma'(x)/\Gamma(x)$ is the digamma function and $\Psi'(x)$ its derivative. This formula in terms of the dimensions (21) generalizes a seminal result of Page [1] that applies to the special case of an a priori factorization of the Hilbert space into subsystems.

It is useful to determine bounds on the average entropy and its variance that do not rely on any special choice of system and subsystem or on any asymptotic limit. We use the following inequalities for the digamma function and its derivative:

$$\log(x) + \tfrac{1}{2x+1} \; < \; \Psi(x+1) \; < \; \log(x) + \tfrac{1}{2x}, \tag{48}$$

$$\tfrac{1}{x} - \tfrac{1}{2x^2+1} \; < \; \Psi'(x+1) \; < \; \tfrac{1}{x} - \tfrac{1}{2x^2}. \tag{49}$$

We start with the average $\langle S \rangle$. For any choice of non-trivial subsystem and for all $r$ we have $b_r < D = \sum_{r'} d_{r'} b_{r'}$. Using (48) we can then put a tight bound on $\varphi_r$:

$$\log\min\left(\tfrac{D}{b_r}, \tfrac{D}{d_r}\right) - \tfrac{1}{2}\min\left(\tfrac{d_r}{b_r}, \tfrac{b_r}{d_r}\right) \; \leq \; \varphi_r \; \leq \; \log\min\left(\tfrac{D}{b_r}, \tfrac{D}{d_r}\right). \tag{50}$$

It follows that the average entropy is bounded from above and from below by

$$\sum_r \frac{d_r b_r}{D}\left(\log\min\left(\tfrac{D}{b_r}, \tfrac{D}{d_r}\right) - \tfrac{1}{2}\min\left(\tfrac{d_r}{b_r}, \tfrac{b_r}{d_r}\right)\right) \; \leq \; \langle S \rangle \; \leq \; \sum_r \frac{d_r b_r}{D}\log\min\left(\tfrac{D}{b_r}, \tfrac{D}{d_r}\right). \tag{51}$$

We note that, as $\min\left(\tfrac{d_r}{b_r}, \tfrac{b_r}{d_r}\right) \leq 1$, we have also the exact inequality

$$\left| \langle S \rangle - \sum_r \frac{d_r b_r}{D}\log\min\left(\tfrac{D}{b_r}, \tfrac{D}{d_r}\right) \right| \leq \tfrac{1}{2}, \tag{52}$$

which is useful for extracting the asymptotics of the average entropy in the limit of large dimension $D$, up to terms of order one. We note also that, because of the min, the upper bound is tighter than the entropy of the average (42), consistently with the inequality $\langle S_{\text{vN}}(\rho_{\mathcal{S}}) \rangle \leq S_{\text{vN}}(\langle \rho_{\mathcal{S}} \rangle)$. Clearly, the average entropy is also smaller than the maximum entropy (37).[4]

We consider then the variance $(\Delta S)^2$. Here we use two (rather loose, but useful) inequalities for the functions $\varphi_r$ and $\chi_r$,

$$0 \leq \varphi_r \leq \log D, \qquad 0 \leq \chi_r \leq 2. \tag{53}$$

---

[4]In fact $\sum_r \frac{d_r b_r}{D}\log\min\left(\tfrac{D}{b_r}, \tfrac{D}{d_r}\right) \leq \log\left(\sum_r \min(d_r, b_r)\right)$ because log is concave and $\langle \log \cdot \rangle \leq \log\langle \cdot \rangle$.

It is immediate then to show that the variance of the entanglement entropy is bounded from above by a decreasing function of the dimension $D$,

$$(\Delta S)^2 \leq \frac{(\log D)^2 + 2}{D}. \tag{54}$$

This bound shows that, independently of the details of the system and of the subsystem, the variance of the entropy vanishes in the large dimension of the Hilbert space limit, i.e., $\Delta S \to 0$ as $D \to \infty$. Therefore, if the average $\langle S \rangle$ is finite and non-vanishing in this limit, then its value represents the typical entanglement entropy of a random pure state restricted to the subsystem.

## 3 Non-Abelian symmetry-resolved states and entropy

We consider a physical system with a symmetry group $G$ that leaves the Hamiltonian of the system invariant. Then the Hilbert space $\mathcal{H}$ of the system carries a (reducible) representation $U$ of the group that is unitary, and $G$-invariant observables $O$ of the system—including the Hamiltonian—satisfy

$$g \in G \quad \Longrightarrow \quad U(g) O U(g)^{-1} = O. \tag{55}$$

Symmetry-resolved states of the system are defined as pure states $|\psi\rangle \in \mathcal{H}$ that remain pure when restricted to the subalgebra of $G$-invariant observables. In this section, we discuss the definition of symmetry-resolved states and the computation of their typical entropy for symmetry-resolved subsystems. Compared to the Abelian case, we show that new features arise for non-Abelian symmetry groups. For concreteness, we focus on semisimple Lie groups such as $G = SU(2)$, i.e., continuous groups that do not have any Abelian invariant subgroup [3, 102, 103]. We then proceed to comment on the general case.

### 3.1 Symmetry-resolved decomposition of the Hilbert space

A compact Lie group $G$ is defined by the Lie algebra of its generators $T^a$ with real structure constants $f^{ab}{}_c$,

$$[T^a, T^b] = \mathrm{i} f^{ab}{}_c T^c. \tag{56}$$

We say that the finite-dimensional Hilbert space $\mathcal{H}$ carries a unitary representation of the group $G$ if the generators $T^a$ are realized as $D \times D$ Hermitian matrices that satisfy the commutation relations (56), with $D = \dim \mathcal{H}$. A group element with real parameters $\alpha_a$ acts on the Hilbert space as the unitary transformation $U$ given by

$$U = \mathrm{e}^{\mathrm{i}\alpha_a T^a}. \tag{57}$$

This representation of the group is, in general, reducible. It is useful to introduce the Cartan-Killing metric $\eta^{ab} = \mathrm{tr}(\tau^a \tau^b)$ where $\tau^a$ are the generators in the adjoint representation, $[\tau^a]^b{}_c = -\mathrm{i} f^{ab}{}_c$. Here we restrict attention to the case of real compact semisimple Lie groups, for which the metric $\eta^{ab}$ is positive definite.[5] We use this metric and its inverse $\eta_{ab}$, with $\eta^{ac}\eta_{cb} = \delta^a{}_b$, to raise and lower indices.

The symmetry generators $T^a$ generate a subalgebra $\mathcal{A}_{\text{sym}}$ of the algebra of observables $\mathcal{A} = \mathcal{L}(\mathcal{H})$ of the system,

$$\mathcal{A}_{\text{sym}} = \mathbb{C}[T^a]. \tag{58}$$

---

[5]Note that in this case, by rescaling the generators, the metric $\eta^{ab}$ can be brought to the Euclidean form $\delta^{ab}$. Here, we use the standard tensorial notation, where we keep track of upper and lower indices, and repeated indices are contracted. As usual, the symbols $^{(\cdots)}$ and $^{[\cdots]}$ stand for complete symmetrization or anti-symmetrization of the tensor.

The commutant of this subalgebra defines the algebra of $G$-invariant observables,

$$\mathcal{A}_G \equiv (\mathcal{A}_{\text{sym}})' = \{M \in \mathcal{A} \,|\, [M, T^a] = 0\}. \tag{59}$$

Observables in $\mathcal{A}_G$ commute with the generators $T^a$ and therefore satisfy the $G$-invariance condition (55).

The rank of a semisimple Lie group $G$ is the dimension rank($G$) of any one of the Cartan subalgebras of its Lie algebra. The number of linearly independent Casimir operators $Q_k$ is exactly given by rank($G$) [104, 105]. The Casimir operators are obtained by listing all the completely-symmetric $G$-invariant tensors $\eta_{a_1 \cdots a_p}$ of order $p$, and then contracting them with the generators:

$$Q_k = \eta_{a_1 \cdots a_{p(k)}} T^{a_1} \cdots T^{a_{p(k)}}, \qquad k = 1, \ldots, \text{rank}(G). \tag{60}$$

These operators generalize the familiar quadratic Casimir operator $Q_1 = \eta_{ab} T^a T^b$ defined in terms of the Cartan-Killing metric. They belong to the $G$-invariant subalgebra $\mathcal{A}_G$ as they are invariants, and they belong to the algebra $\mathcal{A}_{\text{sym}}$ as they are expressed in terms of the generators. Therefore, they belong to the center $\mathcal{Z}_{\text{sym}}$ of the algebra. In fact, for semisimple Lie groups (that is, Lie groups that have no Abelian subgroup), a much stronger result holds [104, 105]: these $R$ linearly independent Casimir operators generate the center,

$$\mathcal{Z}_{\text{sym}} = \mathcal{A}_{\text{sym}} \cap \mathcal{A}_G = \mathbb{C}[Q_k], \tag{61}$$

and characterize completely the irreducible representations of the group.

Using the results of Sec. 2.1, and denoting collectively $q$ the eigenvalues of the Casimir operators, we obtain a symmetry-resolved decomposition of the Hilbert space:

$$\mathcal{H} = \bigoplus_q \left( \mathcal{H}_{\text{sym}}^{(q)} \otimes \mathcal{H}_G^{(q)} \right), \tag{62}$$

where $\mathcal{H}_{\text{sym}}^{(q)}$ carries the irreducible representation $(q)$ of the symmetry genererarators, and $\mathcal{H}_G^{(q)}$ is the space of $G$-invariant degrees of freedom of the system defined as

$$\mathcal{H}_G^{(q)} = \text{Inv}_G (\mathcal{H}_{\text{sym}}^{(\bar{q})} \otimes \mathcal{H}), \tag{63}$$

where $\text{Inv}_G$ denotes the $G$-invariant subspace in the tensor product and $(\bar{q})$ is the conjugate representation [102, 103].

The symmetry-resolved decomposition (62) is a decomposition of the Hilbert space $\mathcal{H}$ into a direct sum of irreducible representations labeled by the quantum numbers $q$ that label the eigenvalues of the Casimir operators. Furthermore, each irreducible representation is a tensor product of the sym factor $\mathcal{H}_{\text{sym}}^{(q)}$ that transforms under an irreducible representation of the group, and the $G$-invariant Hilbert space $\mathcal{H}_G^{(q)}$ of internal degrees of freedom defined by (63). A generic state $|\psi\rangle$ in $\mathcal{H}$ can be expanded on the orthonormal basis adapted to this decomposition:

$$|\psi\rangle = \sum_q \sqrt{p_q} \sum_m \sum_i \psi_{mi}^{(q)} |q, m\rangle_{\text{sym}} |q, i\rangle_G, \tag{64}$$

where $m$ in the basis $|q, m\rangle_{\text{sym}}$ denotes collectively the eigenvalues of the Cartan generators of the group $G$, and $i$ in the basis $|q, i\rangle_G$ labels the internal $G$-invariant degrees of freedom of the system.

We give three examples of the symmetry-resolved Hilbert space for semisimple Lie groups:

$G = SU(2)$ — This is the simplest non-trivial case illustrated in Sec. 1. The generators are the spin operators $\vec{J} = (J^i)$ with $i = 1, 2, 3$, the structure constants are $\epsilon^{ij}{}_k$, the Cartan-Killing metric $\delta_{ij}$. The rank of the group is one, and therefore, the decomposition is in terms of a

single Casimir operator, the quadratic Casimir $Q = \delta_{ik} J^i J^k = \vec{J}^2$. As usual, the eigenvalues of the Casimir operator are written as $j(j+1)$ with $j = 0, \frac{1}{2}, 1, \ldots$ half-integer, and the eigenvalues of the Cartan subalgebra operator $J^z$ are $m = -j, \ldots, +j$. The Hilbert space decomposes as $\mathcal{H} = \bigoplus_j \left( \mathcal{H}_{\text{sym}}^{(j)} \otimes \mathcal{H}_G^{(j)} \right)$, with $\mathcal{H}_{\text{sym}}^{(j)}$ of dimension $\dim \mathcal{H}_{\text{sym}}^{(j)} = 2j + 1$ and orthonormal basis $|j, m\rangle_{\text{sym}}$. In Sec. 5 we discuss a concrete example of symmetry-resolved spin system where the $G$-invariant Hilbert space of internal degrees of freedom $\mathcal{H}_G^{(j)}$ is the space of $SU(2)$ intertwiners of a spin system, which can also be interpreted geometrically as quantum polyhedra [106, 107].

**$G = SU(3)$** — Starting from the $3 \times 3$ Gell-Mann matrices $\lambda^a$ (with $a = 1, \ldots, 8$) and using the textbook normalization $\lambda^a / 2$ of the generators in the fundamental representation [102, 103], we can express the Cartan-Killing metric as $\eta^{ab} = \frac{1}{4} \text{tr}(\lambda^a \lambda^b) = \frac{1}{2} \delta^{ab}$, the structure constants as $f^{abc} = -\frac{i}{4} \text{tr}(\lambda^{[a} \lambda^b \lambda^{c]})$ (which is completely antisymmetric when all indices are raised), and the completely symmetric tensor $d^{abc} = \frac{1}{2} \text{tr}(\lambda^{(a} \lambda^b \lambda^{c)})$. The rank of the group is 2, and therefore, there are two linearly independent Casimir operators $Q_1$ and $Q_2$. The quadratic Casimir operator $Q_1 = \frac{1}{2} \eta_{ab} T^a T^b$ has eigenvalues $\frac{1}{3}(q^2 + p^2 + qp + 3q + 3p)$. The cubic Casimir operator $Q_2 = \frac{1}{8} d_{abc} T^a T^b T^c$ has eigenvalues $\frac{1}{18}(q - p)(q + 2p + 3)(p + 2q + 3)$ sometimes called the anomaly coefficient. The quantum numbers $q, p = 0, 1, 2, \ldots$ label the irreducible representations and the decomposition as a direct sum over the center is $\mathcal{H} = \bigoplus_{q,p} \left( \mathcal{H}_{\text{sym}}^{(q,p)} \otimes \mathcal{H}_G^{(q,p)} \right)$. The dimension of the sym factor is $\dim \mathcal{H}_{\text{sym}}^{(q,p)} = \frac{1}{2}(q + 1)(p + 1)(q + p + 2)$.

**$G = SO(4)$** — The algebra of the group is the same as the algebra of $SU(2)_L \times SU(2)_R$ with generators given by the spin operators $\vec{J}_L$ and $\vec{J}_R$ [3]. The group has rank 2, and the two linearly independent Casimir operators can be taken as $Q_L = \vec{J}_L^2$ and $Q_R = \vec{J}_R^2$. The half-integer quantum numbers $j_L$ and $j_R$ label the irreducible representations, and the symmetry-resolved decomposition of the Hilbert space is $\mathcal{H} = \bigoplus_{j_L, j_R} \left( \mathcal{H}_{\text{sym}}^{(j_L, j_R)} \otimes \mathcal{H}_G^{(j_L, j_R)} \right)$. The dimension of the sym factor is $\dim \mathcal{H}_{\text{sym}}^{(j_L, j_R)} = (2j_L + 1)(2j_R + 1)$.

In general, for the classical compact matrix groups $A_n = SU(n+1)$, $B_n = SO(2n+1)$ and $C_n = Sp(2n)$ of rank $n$, the list of the $n$ linearly independent Casimir operators is given by $Q_k = \eta_{a_1 \ldots a_k} T^{a_1} \cdots T^{a_k}$ with the invariant tensor $\eta_{a_1 \ldots a_k} = \text{tr}(\tau^{(a_1} \cdots \tau^{a_k)})$ defined using where $\tau^a$ in the fundamental representation [105]. If one took $\tau^a$ in the adjoint representation instead, one could not distinguish the representation $(q, p)$ from its conjugate $(p, q)$ in $SU(3)$, for instance. In the case $D_n = SO(2n)$, one can construct the invariant tensors from the spinor representation or, equivalently, one can take the $(n-1)$ Casimir operators of even order, $Q_{2k}$ with $k = 1, \ldots, n-1$, together with the order-$n$ Casimir invariant $\tilde{Q} = \epsilon_{\mu_1 \nu_1 \ldots \mu_n \nu_n} J^{\mu_1 \nu_1} \cdots J^{\mu_n \nu_n}$, where $J^{\mu\nu}$ are the generators. The Casimir operator $\tilde{Q}$ allows us to distinguish the two mirror representations of $SO(2n)$. In the case of $SO(4)$, this construction reduces to the two quadratic invariants $Q = J_{\mu\nu} J^{\mu\nu} = 4(\vec{J}_L^2 + \vec{J}_R^2)$ and $\tilde{Q} = \epsilon_{\mu\nu\rho\sigma} J^{\mu\nu} J^{\rho\sigma} = 8(\vec{J}_L^2 - \vec{J}_R^2)$. A similar construction applies to the exceptional Lie groups $G_2$, $F_4$, $E_6$, $E_7$, $E_8$, with the invariant tensors $\eta_{a_1 \ldots a_k}$ built from the generators in an irreducible representation that is non-degenerate.

Finally, let us comment on the Abelian case using the compact Lie group $U(1)$ or many copies of it:

**$G = U(1) \times \cdots \times U(1)$** — We note that the symmetry-resolved decomposition (62) becomes trivial. In fact, in this case the structure constants $f^{ab}{}_c$ vanish because the generators $T^a$ commute, and therefore their algebra coincides with the center, $\mathcal{A}_{\text{sym}} = \mathcal{Z}_{\text{sym}} = \mathbb{C}[T^a]$. As a result, we have a decomposition of the form $\mathcal{H} = \bigoplus_m \mathcal{H}_G^{(m)}$ where the eigenvalues of the commuting generators $T^a$ (sometimes called charges or particle numbers) are collectively denoted as $m$. The sym factor in the decomposition is trivial, $\dim \mathcal{H}_{\text{sym}}^{(m)} = 1$, and can be reabsorbed into a phase in each sector $\mathcal{H}_G^{(m)}$.

## 3.2 Symmetry-resolved states and subsystems

The symmetry-resolved decomposition of the Hilbert space (62) allows us to write $G$-invariant observables as

$$O \in \mathcal{A}_G \quad \Longrightarrow \quad O = \bigoplus_q \left( \mathbb{1}^{(q)}_{\mathrm{sym}} \otimes O^{(q)}_G \right). \tag{65}$$

Given a pure state $|\psi\rangle \in \mathcal{H}$, we can write the restriction of the state to the $G$-invariant observables as $\rho_G = \langle \psi | \Pi_G | \psi \rangle$, where the map $\Pi_G$ is defined concretely by (26). A *symmetry-resolved state* is defined as a pure state that remains pure when restricted to the $G$-invariant subalgebra of observables $\mathcal{A}_G$. As shown in Sec. 2, Eq. (36), this implies that it belongs to an irreducible representation $q$ and has the product form:

$$|\psi\rangle \quad \text{symmetry-resolved state in } \mathcal{H} \quad \Longleftrightarrow \quad \exists q : \quad |\psi\rangle = |q, \xi\rangle_{\mathrm{sym}} \, |q, \chi\rangle_G. \tag{66}$$

In other words, symmetry-resolved states have no entanglement between internal $G$-invariant degrees of freedom and *sym* degrees of freedom that change under transformations of the group $G$.

A *symmetry-resolved subsystem* is defined by a set $O_1$, $O_2$, ... of $G$-invariant observables that we have access to. The algebra $\mathcal{A}_{G\mathcal{S}}$ that they generate is

$$\mathcal{A}_{G\mathcal{S}} = \mathbb{C}[O_l] \subseteq \mathcal{A}_G, \qquad \text{with} \qquad O_l \in \mathcal{A}_G, \qquad l = 1, \ldots, L. \tag{67}$$

We are interested in the restriction of a state $|\psi\rangle$ to this subalgebra, $\rho_{G\mathcal{S}} = \langle \psi | \Pi_{G\mathcal{S}} | \psi \rangle$. To build the map $\Pi_{G\mathcal{S}}$, we follow the steps discussed in Sec. 2. First, we define the rest of the system using the commutant algebra,

$$\mathcal{A}_{\overline{G\mathcal{S}}} \equiv (\mathcal{A}_{G\mathcal{S}})' = \{ M \in \mathcal{A} \, | \, [M, O_l] = 0, \ \forall \, O_l \in \mathcal{A}_{G\mathcal{S}} \}. \tag{68}$$

Note that $\mathcal{A}_{\overline{G\mathcal{S}}}$ also contains observables that are not $G$-invariant. The center of the subalgebra is generated by a set of commuting $G$-invariant observables $R_i$ in the intersection of the two,

$$\mathcal{Z}_{G\mathcal{S}} = \mathcal{A}_{G\mathcal{S}} \cap \mathcal{A}_{\overline{G\mathcal{S}}} = \mathbb{C}[R_i]. \tag{69}$$

By diagonalizing first the commuting observables $R_i$ and calling collectively their eigenvalues $r$, we obtain the direct sum decomposition

$$\mathcal{H} = \bigoplus_r \left( \mathcal{H}^{(r)}_{G\mathcal{S}} \otimes \mathcal{H}^{(r)}_{\overline{G\mathcal{S}}} \right). \tag{70}$$

This decomposition allows us to define the map $\Pi_{G\mathcal{S}}$. As we are interested in the restriction of a symmetry-resolved state to a symmetry-resolved subsystem, it is useful to have a decomposition of the Hilbert space and an orthonormal basis that is adapted to both decompositions (62) and (70). We show that this is possible in general with a concrete construction.

We introduce a decomposition adapted to symmetry-resolved states and symmetry-resolved subsystems. Let us consider the algebra $\mathcal{A}_{\mathrm{sym}G\mathcal{S}}$ generated by the symmetry generators $T^a$ and by the $G$-invariant observables $O_l$ that generate the subsystem,

$$\mathcal{A}_{\mathrm{sym}G\mathcal{S}} = \mathbb{C}[T^a, O_l]. \tag{71}$$

The commutant of this algebra is[6]

$$\mathcal{A}_{G\,\overline{\mathcal{S}}} \equiv (\mathcal{A}_{\mathrm{sym}G\mathcal{S}})' = \mathbb{C}[T^a, O_l]' = \mathbb{C}[T^a]' \cap \mathbb{C}[O_l]' = (\mathcal{A}_{\mathrm{sym}})' \cap (\mathcal{A}_{G\mathcal{S}})' = \mathcal{A}_G \cap \mathcal{A}_{\overline{G\mathcal{S}}}. \tag{72}$$

---

[6]In the first line we used the relation (15) that applies to any set of observables $H_i$ and $K_j$.

Note that, because of the presence of the symmetry generators $T^a$ in $\mathcal{A}_{\text{sym}GS}$, the algebra $\mathcal{A}_{G\overline{S}}$ contains only the $G$-invariant observables in $\mathcal{A}_{\overline{GS}}$. We can now define the center

$$\mathcal{Z}_{\text{sym}GS} = \mathcal{A}_{\text{sym}GS} \cap \mathcal{A}_{G\overline{S}} = \mathbb{C}[Q_k, R_i]. \tag{73}$$

Note that the center of the algebra $\mathcal{A}_{\text{sym}GS}$ is generated by the elements of the center $R_i$ of the symmetry-resolved subsystem and by the Casimir operators $Q_k$ of the group $G$. The fact that the two commutes, $[R_i, Q_k] = 0$, follows immediately from the fact that $R_i$ are $G$-invariant observables, i.e., $[R_i, T^a] = 0$, and the Casimir operators (60) are functions of the symmetry generator. By diagonalizing simultaneously the commuting set $\{Q_k, R_i\}$ with eigenvalues denoted collectively by $q$ and $r$, we obtain the decomposition

$$\mathcal{H} = \bigoplus_q \Big( \mathcal{H}_{\text{sym}}^{(q)} \otimes \bigoplus_r \big( \mathcal{H}_{GS}^{(r)} \otimes \mathcal{H}_{G\overline{S}}^{(q,r)} \big) \Big). \tag{74}$$

The decomposition comes with an orthonormal basis adapted simultaneously to symmetry-resolved states and to the symmetry-resolved subsystems,

$$|q, r, \alpha, \beta\rangle = |q, m\rangle_{\text{sym}} |r, \alpha\rangle_{GS} |q, r, \beta\rangle_{G\overline{S}}. \tag{75}$$

Note that the basis elements $|r, \alpha\rangle_{GS}$ of the symmetry-resolved system depend on the eigenvalues $r$ of $R_i$ but not on the eigenvalues $q$ of the Casimirs $Q_k$. This feature can also be understood by considering the two decompositions $\mathcal{H}_G^{(q)} = \bigoplus_r \big( \mathcal{H}_{GS}^{(r)} \otimes \mathcal{H}_{G\overline{S}}^{(q,r)} \big)$ and $\mathcal{H}_{G\overline{S}}^{(r)} = \bigoplus_q \big( \mathcal{H}_{\text{sym}}^{(q)} \otimes \mathcal{H}_{G\overline{S}}^{(q,r)} \big)$, that relate the formulas (62), (70) and (74).

### 3.3 Symmetry-resolved entanglement entropy

The entanglement entropy of a symmetry-resolved state $|\psi\rangle$ restricted to a symmetry-resolved subsystem can be defined and computed using the tools introduced above. The symmetry-resolved state can be first written in the basis adapted to the subsystem:

$$|\psi\rangle = \Big( \sum_m \xi_m |q, m\rangle_{\text{sym}} \Big) \Big( \sum_r \sqrt{p_r} \sum_{\alpha, \beta} \chi_{\alpha\beta}^{(r)} |r, \alpha\rangle_{GS} |q, r, \beta\rangle_{G\overline{S}} \Big), \tag{76}$$

with $p_r$, the probability of finding the state in the sector is $r$. The state belongs to a definite sector $q$, it does not have entanglement between internal $G$-invariant degrees of freedom and sym degrees of freedom, and, in general, can have entanglement between the $G$-invariant degrees of freedom in the subsystem and its complement. The restriction to the subsystem can be written as

$$\rho_{GS} = \langle \psi | \Pi_{GS} | \psi \rangle = \bigoplus_r p_r \, \rho_{GS}^{(r)}, \qquad \text{with} \qquad (\rho_{GS}^{(r)})_{\alpha\alpha'} = \sum_\beta \chi_{\alpha\beta}^{(r)} \chi_{\alpha'\beta}^{(r)*}. \tag{77}$$

The entanglement entropy $S(|\psi\rangle, \mathcal{A}, \mathcal{A}_{GS})$ can then be computed using (29), (35). Bounds on the entanglement entropy can be written in terms of the dimensions of the sectors:

$$d_r = \dim \mathcal{H}_{GS}^{(r)}, \qquad b_{qr} = \dim \mathcal{H}_{G\overline{S}}^{(q,r)}, \qquad D_q = \dim \mathcal{H}_G^{(q)} = \sum_r d_r b_{qr}. \tag{78}$$

In particular a symmetry-resolved state $|\psi^{(q)}\rangle$ in the sector $q$ has entanglement entropy bounded from above by

$$0 \leq S(|\psi^{(q)}\rangle, \mathcal{A}, \mathcal{A}_{GS}) \leq \log \Big( \sum_r \min(d_r, b_{qr}) \Big). \tag{79}$$

Note that here, the sector $\mathcal{H}_{\text{sym}}^{(q)}$ is simply an ancilla, and the entropy is independent of $\xi_m$.

A random symmetry-resolved state with fixed $q$ can be defined starting from a reference state of the symmetry-resolved form $|q, \xi\rangle_{\text{sym}} |q, \chi\rangle_G$, and acting on it with a random unitary of the form $\bigoplus_q (U_{\text{sym}}^{(q)} \otimes U_G^{(q)})$, where $U_{\text{sym}}^{(q)}$ and $U_G^{(q)}$ are Haar-measure distributed on $\mathcal{H}_{\text{sym}}^{(q)}$ and $\mathcal{H}_G^{(q)}$ respectively. The average entanglement entropy and its variance are then given by the expressions (43) and (44), with the dimensions $d_r$ and $b_{qr}$ given above.

# 4 Locality, many-body systems and $G$-local entanglement

In a lattice many-body system, there is a built-in notion of locality associated with the $N$ bodies, or particles, at the sites of the lattice. We assume that the Hilbert space at each site $n$ is a copy of a finite-dimensional Hilbert space $\mathcal{H}_d \simeq \mathbb{C}^d$ that carries a unitary (reducible) representation of a compact Lie group $G$. Therefore, the kinematical Hilbert space $\mathcal{H}_N$ of the system is the tensor product of the Hilbert spaces at sites [18, 108]:

$$\mathcal{H}_N = \underbrace{\mathcal{H}_d \otimes \cdots \otimes \mathcal{H}_d}_{N} . \tag{80}$$

Calling $T_n^a$ the $d$-dimensional (reducible) representation of the generators of the group $G$ at each site $n$, we have that the generator of global transformation for the group $G$ acting on $\mathcal{H}_N$ is simply given by the sum

$$T^a = \sum_{n=1}^{N} T_n^a . \tag{81}$$

We can then use the results of Sec. 3, and in particular (62)–(63), to decompose the Hilbert space in symmetry-resolved sectors

$$\mathcal{H}_N = \bigoplus_q \mathcal{H}_N^{(q)} = \bigoplus_q \left( \mathcal{H}_{\text{sym}}^{(q)} \otimes \mathcal{H}_G^{(q)} \right), \tag{82}$$

where the quantum number $q$ labels the irreducible representations of the group $G$, i.e., the eigenvalues of the Casimir operators for a semisimple Lie group. The representation space $\mathcal{H}_{\text{sym}}^{(q)}$ is the one already described in Sec. 3.1 and the invariant space is

$$\mathcal{H}_G^{(q)} = \text{Inv}_G \left( \mathcal{H}_{\text{sym}}^{(\bar{q})} \otimes \underbrace{\mathcal{H}_d \otimes \cdots \otimes \mathcal{H}_d}_{N} \right), \tag{83}$$

where $\bar{q}$ is the conjugate representation. In this section, we discuss how the new ingredient of locality associated with the many-body system allows us to define the distinct notions of $K$-local and $G$-local observables illustrated in the introduction in Fig. (1),(2),(3).

## 4.1 $K$-local decomposition of the Hilbert space

Let us consider a subset $n \in A$ of the nodes of the many-body lattice, for instance, the ones defining a local region $A$ of the lattice and excluding its complement $B$ [108]. This subsystem corresponds to the standard tensor-product decomposition

$$\mathcal{H}_N = \mathcal{H}_A \otimes \mathcal{H}_B , \qquad \text{with} \qquad \mathcal{H}_A = \bigotimes_{n \in A} \mathcal{H}_d , \qquad \mathcal{H}_B = \bigotimes_{n \in B} \mathcal{H}_d . \tag{84}$$

This decomposition is associated with a $K$-local subalgebra of observables. Let us define first the kinematical algebra of observables $\mathcal{A}_K = \mathcal{L}(\mathcal{H}_N)$. The $K$-local subalgebra in $A$ is

$$\mathcal{A}_{KA} = \{ O \in \mathcal{A}_K \mid O = O_A \otimes \mathbb{1}_B \ \text{with} \ O_A \in \mathcal{L}(\mathcal{H}_A) \} . \tag{85}$$

Clearly we have that the $K$-local subalgebra in $B$ coincides with the commutant of $A$, i.e., $\mathcal{A}_{KB} = (\mathcal{A}_{KA})'$, and that the center is trivial $\mathcal{A}_{KA} \cap \mathcal{A}_{KB} = \mathbb{1}$.

We note that the operator (81) that generates global unitary transformations for the group $G$ takes the form

$$T^a = T_A^a \otimes \mathbb{1}_B + \mathbb{1}_A \otimes T_B^a, \tag{86}$$

which is additive over the subsystems $A$ and $B$. The operator $T_A^a$ given by

$$T_A^a \otimes \mathbb{1}_B = \sum_{n \in A} T_n^a \in \mathcal{A}_{KA}, \tag{87}$$

and generates unitary transformations for the group $G$ in $A$.[7]

## 4.2 $G$-local decomposition and symmetry-resolution

$G$-invariant observables $O_G$ of the many-body system belong to the algebra $\mathcal{A}_G$ defined in (59). $G$-local observables in the subsystem $A$ are observables that are both $G$-invariant and belong to the subsystem $A$. They are, therefore, elements of the intersection of the two algebras,

$$\mathcal{A}_{GA} = \mathcal{A}_{KA} \cap \mathcal{A}_G. \tag{88}$$

We can similarly define $G$-local observables in $B$,

$$\mathcal{A}_{GB} = \mathcal{A}_{KB} \cap \mathcal{A}_G. \tag{89}$$

Note that in general, for a non-Abelian group, the two subalgebras are not the commutant of each other, $\mathcal{A}_{GB} \neq (\mathcal{A}_{GA})'$. In fact, using $\mathcal{A}_G = (\mathcal{A}_{\text{sym}})'$ and the intersection formula (15), we find

$$(\mathcal{A}_{GA})' = (\mathcal{A}_{KA} \cap \mathcal{A}_G)' = \big((\mathcal{A}_{KB})' \cap (\mathcal{A}_{\text{sym}})'\big)' = \mathbb{C}[\mathbb{1}_A \otimes O_B, T_A^a \otimes \mathbb{1}_B]. \tag{90}$$

We can now determine the center of the subalgebra,

$$\mathcal{Z}_{GA} = \mathcal{A}_{GA} \cap (\mathcal{A}_{GA})' = \mathbb{C}\big[Q_A^{(1)}, \ldots, Q_A^{(\text{rank}(G))}\big], \tag{91}$$

which is generated by the Casimir operators in the subsystem $A$,

$$Q_A^{(k)} = \eta_{a_1 \cdots a_{p(k)}} T_A^{a_1} \cdots T_A^{a_{p(k)}}, \qquad k = 1, \ldots, \text{rank}(G), \tag{92}$$

with the symmetric tensors $\eta_{a_1 \cdots a_{p(k)}}$ defined in (60). Denoting collectively $q_A$ the eigenvalues of the subsystem Casimir operators in $A$, and using the results of Sec. 3.2, we obtain a decomposition of the Hilbert space sector $\mathcal{H}_N^{(q)}$ in $G$-local subsystems:

$$\mathcal{H}_N^{(q)} = \mathcal{H}_{\text{sym}}^{(q)} \otimes \bigoplus_{q_A} \big(\mathcal{H}_{GA}^{(q_A)} \otimes \mathcal{H}_{GB}^{(q, q_A)}\big), \tag{93}$$

where the factors are defined as

$$\mathcal{H}_{GA}^{(q_A)} = \text{Inv}_G(\mathcal{H}_{\text{sym}}^{(\bar{q}_A)} \otimes \mathcal{H}_A), \qquad \mathcal{H}_{GB}^{(q, q_A)} = \text{Inv}_G(\mathcal{H}_{\text{sym}}^{(\bar{q})} \otimes \mathcal{H}_{\text{sym}}^{(q_A)} \otimes \mathcal{H}_B). \tag{94}$$

This decomposition provides us with an orthonormal basis of $\mathcal{H}_N^{(q)}$ adapted to the subalgebra $\mathcal{A}_{GA}$. It is then immediate to compute the entanglement entropy $S_{GA}$ of a state $|\psi\rangle$ restricted to the $G$-local subalgebra in $A$,

$$S_{GA} = S(|\psi\rangle, \mathcal{A}, \mathcal{A}_{GA}). \tag{95}$$

---

[7]Note that we are using the same notation for the generator acting on the Hilbert space of a single site $T_n^a \in \mathcal{L}(\mathcal{H}_d)$ and for the generator acting on a single site of the many-body Hilbert space, $T_n^a \equiv \mathbb{1}_d \otimes \cdots \otimes T_n^a \otimes \cdots \otimes \mathbb{1}_d \in \mathcal{L}(\mathcal{H}_d \otimes \cdots \otimes \mathcal{H}_d)$.

It is interesting to compare the properties of the $G$-local entanglement entropy $S_{GA}$ to the ones of the familiar $K$-local entanglement entropy $S_{KA}$,

$$S_{KA} = S(|\psi\rangle, \mathcal{A}, \mathcal{A}_{KA}). \tag{96}$$

In general, the two entropies do not coincide because the $G$-local subalgebra can be understood as a coarse-graining of the $K$-local one [29], i.e., $\mathcal{A}_{GA} \subset \mathcal{A}_{KA}$. They both probe local properties of the many-body system and can be understood as functions of the number of bodies $N_A$ in the subsystem $A$. However, there is a crucial difference between the two. Commutant symmetry (39) implies that $S_{KA}(|\psi\rangle) = S_{KB}(|\psi\rangle)$, but in general $S_{GA}(|\psi\rangle) \neq S_{GB}(|\psi\rangle)$ because $\mathcal{A}_{GB} \neq (\mathcal{A}_{GA})'$.

Bringing together the results of Sec. 2, Sec. 3, and the decomposition (93), we obtain the exact formulas for the typical $G$-local symmetry-resolved entanglement entropy that apply to a compact Lie group $G$. For a random symmetry-resolved state $|\psi^{(q)}\rangle$ with total charge $q$, the total symmetry-resolved entanglement entropy has average value

$$\langle S_{GA}\rangle_q = \sum_{q_A|\, d_{q_A}\leq b_{qq_A}} \frac{d_{q_A} b_{qq_A}}{D}\left(\Psi(D+1) - \Psi(b_{qq_A}+1) - \frac{d_{q_A}-1}{2\,b_{qq_A}}\right) \tag{97}$$

$$+ \sum_{q_A|\, d_{q_A}>b_{qq_A}} \frac{d_{q_A} b_{qq_A}}{D}\left(\Psi(D+1) - \Psi(d_{q_A}+1) - \frac{b_{qq_A}-1}{2\,d_{q_A}}\right), \tag{98}$$

where the dimensions of the sectors are defined as

$$d_{q_A} = \mathcal{H}_{GA}^{(q_A)}, \qquad b_{qq_A} = \dim \mathcal{H}_{GB}^{(q,q_A)}, \qquad D = \dim \mathcal{H}_{GA}^{(q_A)} = \sum_{q_A} d_{q_A} b_{qq_A}. \tag{99}$$

Note that the formula takes into account the fact that, in the sum over subsystem charges $q_A$, the dimensions of the subsystem sectors can satisfy either $d_{q_A} \leq b_{qq_A}$ or $d_{q_A} > b_{qq_A}$.

Following the definitions in table 10, we can also decompose the total symmetry-resolved entanglement entropy into different components. The average symmetry-resolved entanglement entropy at fixed subsystem charge, $\langle S_{GA}^{(q_A)}\rangle_q$, i.e., the entanglement entropy of a random symmetry-resolved state $|\psi^{(q)}\rangle$ projected to the sector with subsystem charge $q_A$, is given by the standard Page formula [1,2]

$$\langle S_{GA}^{(q_A)}\rangle_q = \begin{cases} \Psi(d_{q_A} b_{qq_A} + 1) - \Psi(b_{qq_A}+1) - \dfrac{d_{q_A}-1}{2\,b_{qq_A}}, & d_{q_A} \leq b_{qq_A}, \\[2ex] b_{qq_A} \longleftrightarrow d_{q_A}, & d_{q_A} > b_{qq_A}. \end{cases} \tag{100}$$

The average over the subsystem sectors $q_A$, weighted with the probability,

$$\varrho_{q_A} = \frac{d_{q_A} b_{qq_A}}{D}, \tag{101}$$

of the state $|\psi^{(q)}\rangle$ being found in each sector, is given by the configurational entropy

$$\langle S_{GA}^{(\mathrm{conf})}\rangle_q = \sum_{q_A} \varrho_{q_A} \langle S_{GA}^{(q_A)}\rangle_q. \tag{102}$$

The average Shannon entropy of the probability distribution (101) defines the number entropy

$$\langle S_{GA}^{(\mathrm{num})}\rangle_q = \sum_{q_A} \varrho_{q_A} \left(\Psi(D+1) - \Psi(d_{q_A} b_{qq_A} + 1)\right). \tag{103}$$

Finally, the total symmetry-resolved entanglement entropy can be written as the sum

$$\langle S_{GA}\rangle_q = \langle S_{GA}^{(\mathrm{conf})}\rangle_q + \langle S_{GA}^{(\mathrm{num})}\rangle_q. \tag{104}$$

### 4.3 $U(1)$ symmetry-resolved entanglement

In the case of an Abelian symmetry group there are significant simplifications that we illustrate here with the example of $G = U(1)$ and charge conservation [4–12].

We consider a lattice many-body system with nodes carrying a copy of the two-dimensional Hilbert space $\mathcal{H}_2 = \mathbb{C}^2$ of a spin-$1/2$ particle, and transforming under a reducible representation of the group $U(1)$, i.e.,

$$\mathcal{H}_N = \underbrace{\mathcal{H}_2 \otimes \cdots \otimes \mathcal{H}_2}_{N}, \qquad \text{with} \qquad \mathcal{H}_2 = \bigoplus_{m'=\pm 1/2} \mathcal{H}_2^{(m')} = \text{span}(|1/2, \pm 1/2\rangle). \tag{105}$$

The generator of the $U(1)$ symmetry at each site is the spin operator $S_n^z = \sigma^z/2$, and the generator of global $U(1)$ transformations is

$$J^z = \sum_{n=1}^{N} S_n^z, \tag{106}$$

which generates global rotations that preserve the direction of the axis $z$. The Hilbert space of the system decomposes as a direct sum over irreducible representations of the group $U(1)$ labeled by the quantum number $m$, the eigenvalue of $J^z$ or the total charge,

$$\mathcal{H}_N = \bigoplus_{m=-N/2}^{+N/2} \left(\mathcal{H}_{\text{sym}}^{(m)} \otimes \mathcal{H}_G^{(m)}\right) = \bigoplus_{m=-N/2}^{+N/2} \mathcal{H}_G^{(m)}. \tag{107}$$

We note that, as the group is Abelian, the sym factor $\mathcal{H}_{\text{sym}}^{(m)}$ is trivial, it has dimension $\dim \mathcal{H}_{\text{sym}}^{(m)} = 1$ and the pure phase $e^{im\theta}$ associated to each irreducible representation $m$ can be reabsorbed in the $U(1)$-invariant part of the state in $\mathcal{H}_G^{(m)}$. Moreover, as for $U(1)$ the conjugate representation is simply $\overline{m} = -m$, the invariant Hilbert space is given by

$$\mathcal{H}_G^{(m)} = \text{Inv}_G\left(\mathcal{H}_{\text{sym}}^{(-m)} \otimes \mathcal{H}_N\right) = \bigoplus_{\substack{m_1,\dots,m_N=\pm 1/2 \\ m_1+\cdots+m_N=m}} \mathcal{H}_2^{(m_1)} \otimes \cdots \otimes \mathcal{H}_2^{(m_N)}. \tag{108}$$

To define a $G$-local subsystem with $N_A$ bodies, we define the generator of $U(1)$ transformations in $A$,

$$J_A^z = \sum_{n\in A} S_n^z, \tag{109}$$

with eigenvalues $m_A$. The decomposition in $G$-local subsystems is then given by

$$\mathcal{H}_G^{(m)} = \bigoplus_{m_A}\left(\mathcal{H}_{GA}^{(m_A)} \otimes \mathcal{H}_{GB}^{(m,m_A)}\right) = \bigoplus_{\substack{m_A,m_B \\ m_A+m_B=m}}\left(\mathcal{H}_{GA}^{(m_A)} \otimes \mathcal{H}_{GB}^{(m_B)}\right), \tag{110}$$

where we used the relation $\mathcal{H}_{\text{sym}}^{(m_A)} \otimes \mathcal{H}_{\text{sym}}^{(m_B)} = \mathcal{H}_{\text{sym}}^{(m_A+m_B)}$ that holds only in the Abelian case, together with the definitions

$$\mathcal{H}_{GA}^{(m_A)} \equiv \text{Inv}_G(\mathcal{H}_{\text{sym}}^{(-m_A)} \otimes \mathcal{H}_A), \tag{111}$$

$$\mathcal{H}_{GB}^{(m,m_A)} \equiv \text{Inv}_G(\mathcal{H}_{\text{sym}}^{(-m)} \otimes \mathcal{H}_{\text{sym}}^{(m_A)} \otimes \mathcal{H}_B) = \text{Inv}_G(\mathcal{H}_{\text{sym}}^{(-m+m_A)} \otimes \mathcal{H}_B) = \mathcal{H}_{GB}^{(m-m_A)}. \tag{112}$$

Note that, in the Abelian case, for a symmetry-resolved state $|\psi^{(m)}\rangle \in \mathcal{H}_G^{(m)}$, the $G$-local and the $K$-local entropy coincide,

$$G = U(1) \implies S_{GA}(|\psi^{(m)}\rangle) = S_{KA}(|\psi^{(m)}\rangle), \tag{113}$$

and the commutant symmetry (39) implies the subsystem symmetry

$$G = U(1) \quad \Longrightarrow \quad S_{GB}(|\psi^{(m)}\rangle) = S_{GA}(|\psi^{(m)}\rangle). \tag{114}$$

Applying the general formulas (97)–(104) to the Abelian Lie group $G = U(1)$, we see that the total symmetry-resolved entanglement entropy in the sector of charge $m$ has the average value

$$\langle S_{GA} \rangle_m = \sum_{m_A \mid d_{m_A} \leq b_{m m_A}} \frac{d_{m_A} b_{m m_A}}{D} \left( \Psi(D+1) - \Psi(b_{m m_A}+1) - \frac{d_{m_A}-1}{2 b_{m m_A}} \right) \tag{115}$$

$$+ \sum_{m_A \mid d_{m_A} > b_{m m_A}} \frac{d_{m_A} b_{m m_A}}{D} \left( \Psi(D+1) - \Psi(d_{m_A}+1) - \frac{b_{m m_A}-1}{2 d_{m_A}} \right), \tag{116}$$

where the dimensions of the sectors are given by

$$D_m = \dim \mathcal{H}_G^{(m)} = \binom{N}{\frac{N}{2}+m}, \tag{117}$$

$$d_{m_A} = \dim \mathcal{H}_{GA}^{(m_A)} = \binom{N_A}{\frac{N_A}{2}+m_A}, \tag{118}$$

$$b_{m m_A} = \dim \mathcal{H}_{GB}^{(m,m_A)} = \binom{N-N_A}{\frac{N-N_A}{2}+m-m_A}. \tag{119}$$

Note that the formula takes into account the fact that, in the sum over subsystem charges $q_A$, the dimensions of the subsystem sectors can satisfy either $d_{m_A} \leq b_{m m_A}$ or $d_{m_A} > b_{m m_A}$.

Following the definitions in table 10, we can also decompose the total symmetry-resolved entanglement entropy into different components. The average symmetry-resolved entanglement entropy at fixed subsystem charge, $\langle S_{GA}^{(m_A)} \rangle_m$, i.e., the entanglement entropy of a random symmetry-resolved state $|\psi^{(m)}\rangle$ projected to the sector with subsystem charge $m_A$, is given by the standard Page formula [1,2]

$$\langle S_{GA}^{(m_A)} \rangle_m = \begin{cases} \Psi(d_{m_A} b_{m m_A}+1) - \Psi(b_{m m_A}+1) - \dfrac{d_{m_A}-1}{2 b_{m m_A}}, & d_{m_A} \leq b_{m m_A}, \\ b_{m m_A} \longleftrightarrow d_{m_A}, & d_{m_A} > b_{m m_A}. \end{cases} \tag{120}$$

The average over the subsystem sectors $m_A$, weighted with the probability,

$$\varrho_{m_A} = \frac{d_{m_A} b_{m m_A}}{D}, \tag{121}$$

of the state $|\psi^{(m)}\rangle$ being found in each sector, is given by the configurational entropy

$$\langle S_{GA}^{(\text{conf})} \rangle_m = \sum_{m_A} \varrho_{m_A} \langle S_{GA}^{(m_A)} \rangle_m. \tag{122}$$

The average Shannon entropy of the probability distribution (101) defines the number entropy

$$\langle S_{GA}^{(\text{num})} \rangle_m = \sum_{m_A} \varrho_{m_A} \Big( \Psi(D+1) - \Psi(d_{m_A} b_{m m_A}+1) \Big), \tag{123}$$

Finally, the total symmetry-resolved entanglement entropy can be written as the sum

$$\langle S_{GA} \rangle_m = \langle S_{GA}^{(\text{conf})} \rangle_m + \langle S_{GA}^{(\text{num})} \rangle_m. \tag{124}$$

To compare to the literature on the thermodynamic limit [4–8] and on equipartition of entanglement [15], we introduce intensive quantities and study the behavior of subsystem entropy in the limit $N \to \infty$ at fixed intensive properties. Specifically, we define the subsystem fraction $f$, the system $U(1)$ charge density $s$, and the subsystem charge density $t$ as follows:

$$f = \frac{N_A}{N}, \qquad s = \frac{2m}{N}, \qquad t = \frac{2m_A}{N_A}, \tag{125}$$

and we restrict here to $f < 1/2$. At the leading order in $N$, the symmetry-resolved entanglement entropy at fixed system and subsystem charge reduces to

$$\langle S_{GA}^{(m_A)} \rangle_m = \log d_{m_A} + O(1) = fN\beta(t) - \frac{1}{2}\log N + O(1), \tag{126}$$

where we used the property (48) that allows us to write the digamma function $\Psi(x)$ as a logarithm at the leading order, and we have defined the function $\beta(t)$ as:

$$\beta(t) = -\frac{1-t}{2}\log\left(\frac{1-t}{2}\right) - \frac{1+t}{2}\log\left(\frac{1+t}{2}\right). \tag{127}$$

This result corresponds exactly to the one found in [6].[8] We then compute the configurational entropy, number entropy, and total symmetry-resolved entropy in the thermodynamic limit. In this limit, the probability $\varrho_{m_A}$ (121) is approximated by a discrete Gaussian probability with mean $\bar{m}_A = Nf\frac{s}{2}$ and variance $\sigma_A^2 = N\frac{f(1-f)}{4}$. In terms of intensive quantities, this translates into a continuous probability density function $\varrho(t)$ with mean $\bar{t} = s$ and variance $\sigma_t^2 = \frac{1-f}{fN}$. We evaluate the configurational entropy at leading order in $N$ using saddle-point techniques. This is equivalent to computing the symmetry-resolved entanglement entropy at fixed system and subsystem charge (126) at the mean $t = \bar{t} = s$:

$$\langle S_{GA}^{(\text{conf})} \rangle_m = fN\beta(s) - \frac{1}{2}\log N + O(1). \tag{128}$$

The computation of the number entropy is straightforward. The Shannon entropy of a discrete Gaussian probability is simply given by $\frac{1}{2}\log(2\pi\sigma_A^2) + \frac{1}{2}$ where $\sigma_A$ is the variance. Hence, the number entropy is:

$$\langle S_{GA}^{(\text{num})} \rangle_m = \frac{1}{2}\log\left(N\frac{\pi f(1-f)}{2}\right) + \frac{1}{2} + O(1) = \frac{1}{2}\log N + O(1). \tag{129}$$

We note that when we sum the configurational entropy and the number entropy to obtain the total symmetry-resolved entanglement entropy, the logarithmic contributions cancel, resulting in the formula

$$\langle S_{GA} \rangle_m = fN\beta(s) + O(1), \tag{130}$$

with no logarithmic corrections.

Finally, we comment on the equipartition (or lack of equipartition) of entanglement entropy in the thermodynamic limit, as discussed in [15] and [6]. By equipartition of entanglement, one means that the entropy $\langle S_{GA}^{(m_A)} \rangle_m$ is independent of $m_A$ in some limit. As we found that $\langle S_{GA}^{(m_A)} \rangle_m \approx fN\beta(t)$ with $t = 2m_A/N_A$, we conclude that there is no equipartition of entanglement entropy, as the leading order in $N$ depends explicitly on the subsystem charge $m_A$, as already found in [6]. Furthermore, following the argument in [6], we emphasize the importance of the order of limits. If $m_A$ is fixed before taking the limit $N \to \infty$, using the expansion $\beta(t) = \log 2 - \frac{1}{2}t^2 + O(t^4)$, we obtain instead $\langle S_{GA}^{(m_A)} \rangle_m \approx fN\log 2$. This result matches that of [15], where the leading order is independent of $m_A$ and the equipartition of entanglement entropy is restored.

---

[8]A note about conventions. In [6], the $U(1)$ charges are defined as positive quantities. To compare the formulas, one can use the relations $M \equiv m + N/2$ and $Q = m_A + N_A/2$.

# 5 $SU(2)$ symmetry-resolved entanglement in a spin system

We consider a system consisting of $N$ spin-$\frac{1}{2}$ particles. Each particle has Hilbert space $\mathcal{H}^{(1/2)} \simeq \mathbb{C}^2$, and the Hilbert space of the system comes with a built-in tensor product structure,

$$\mathcal{H}_N = \underbrace{\mathcal{H}^{(1/2)} \otimes \cdots \otimes \mathcal{H}^{(1/2)}}_{N}. \tag{131}$$

The spin operators $\vec{S}_n = (S_n^x, S_n^y, S_n^z)$ generate $SU(2)$ rotations of each particle, satisfy the algebra

$$[S_n^i, S_{n'}^j] = \mathrm{i}\,\delta_{nn'}\,\epsilon^{ij}{}_k\,S_n^k, \qquad n, n' = 1, \ldots, N, \tag{132}$$

and can be represented in terms of Pauli matrices $\vec{\sigma}$ as $\vec{S}_n = \mathbb{1}_2 \otimes \ldots \otimes \frac{\vec{\sigma}}{2} \otimes \ldots \otimes \mathbb{1}_2$. We also introduce the total spin operator $\vec{J}$,

$$\vec{J} = \sum_{n=1}^{N} \vec{S}_n, \tag{133}$$

which generates $SU(2)$ rotations of the full system, i.e., $[J^i, S_n^j] = \mathrm{i}\,\epsilon^{ij}{}_k\,S_n^k$. As usual, we write the eigenvalues of $\vec{J}^2$ as $j(j+1)$. There is a minimum and a maximum spin of the system, which depends on the number $N$ of particles:

$$N \text{ even} \implies j_{\min} = 0, \qquad j_{\max} = \frac{N}{2}, \qquad j \text{ integer}, \tag{134}$$

$$N \text{ odd} \implies j_{\min} = \frac{1}{2}, \qquad j_{\max} = \frac{N}{2}, \qquad j \text{ half-integer}. \tag{135}$$

Following the logic discussed in Sec. 3, we can decompose the Hilbert space $\mathcal{H}_N$ of the system into a direct sum of $SU(2)$ symmetry-resolved sectors. This decomposition allows us to define symmetry-resolved states. Moreover, using the techniques of Sec. 4, we can introduce a notion of $G$-local subsystem for the non-Abelian group $G = SU(2)$.

## 5.1 Symmetry-resolved decomposition of the Hilbert space

We start with the algebra of observables of the system, which we call the *kinematical* algebra $\mathcal{A}_K$ to distinguish it from the group-invariant algebra $\mathcal{A}_G$ discussed later. The kinematical algebra of observables of the system is generated by the spin operators $S_n^i$,

$$\mathcal{A}_K = \mathcal{L}(\mathcal{H}_N) = \mathbb{C}[S_n^i], \qquad \text{with} \qquad n = 1, \ldots, N, \tag{136}$$

where $\mathcal{L}(\mathcal{H}_N) \simeq M_{2^N}(\mathbb{C})$ is the set of linear operators on $\mathcal{H}_N$, and $\mathbb{C}[S_n^i]$ denotes the algebra generated by $S_n^i$ as defined in (13). The subalgebra generated by the $SU(2)$ *symmetry generators* $J^i$ is

$$\mathcal{A}_{\mathrm{sym}} = \mathbb{C}[J^i]. \tag{137}$$

We can introduce then the commutant $(\mathcal{A}_{\mathrm{sym}})'$, which defines the algebra $\mathcal{A}_G$ of group-invariant observables,

$$\mathcal{A}_G \equiv (\mathcal{A}_{\mathrm{sym}})' = \{M \in \mathcal{A}_K \mid [M, J^i] = 0\} = \mathbb{C}[\vec{S}_n \cdot \vec{S}_{n'}]. \tag{138}$$

This is the algebra of observables that commute with the symmetry generators $J^i$ or, equivalently, that are invariant under rotations:

$$O \in \mathcal{A}_G \iff U O U^{-1} = O, \qquad \text{with} \qquad U = \mathrm{e}^{\mathrm{i}\alpha_i J^i}. \tag{139}$$

The center of the subalgebra is generated by the Casimir operator $\vec{J}^2$,

$$\mathcal{Z}_{\mathrm{sym}} = \mathcal{A}_{\mathrm{sym}} \cap \mathcal{A}_G = \mathbb{C}[\vec{J}^2]. \tag{140}$$

The symmetry-resolved decomposition of the Hilbert space is then given by a direct sum over the eigenvalues of the elements in the center,

$$\mathcal{H}_N = \bigoplus_{j=j_{\min}}^{j_{\max}} \left( \mathcal{H}_{\text{sym}}^{(j)} \otimes \mathcal{H}_G^{(j)} \right), \tag{141}$$

that is a sum over the irreducible representations $j$, with each $j$-sector consisting of a tensor product of the Hilbert spaces for *rotational symmetry* degrees of freedom and for *internal* (rotationally invariant) degrees of freedom. The rotational-symmetry degrees of freedom span a Hilbert space of dimension $\dim \mathcal{H}_{\text{sym}}^{(j)} = 2j + 1$, with an orthonormal basis given by the spin-$j$ states

$$|j, m\rangle \in \mathcal{H}_{\text{sym}}^{(j)}, \tag{142}$$

where $m = -j, \ldots, +j$ are the eigenvalues of $J^z$. The internal degrees of freedom can be understood as the $SU(2)$-invariant tensors in the tensor product of $N$ spin-$\frac{1}{2}$ representations and one spin-$j$ representation, which form the intertwiner space

$$\mathcal{H}_G^{(j)} = \text{Inv}_G \left( \mathcal{H}^{(j)} \otimes \underbrace{\mathcal{H}^{(1/2)} \otimes \cdots \otimes \mathcal{H}^{(1/2)}}_{N} \right). \tag{143}$$

Note that we have used Eq. (83) with the conjugate representation $\bar{j} = j$ for the group $SU(2)$. An orthonormal basis of this space is given by the recoupling basis [23],

$$|j, k_1, \ldots, k_{N-2}\rangle = \quad \begin{array}{c} \text{(diagram)} \end{array} \quad \in \mathcal{H}_G^{(j)}. \tag{144}$$

The quantum numbers $k_1, \ldots, k_{N-2}$ label the eigenvalues $k_r(k_r + 1)$ of the recoupling operators $\vec{K}_r^2$. These operators form a maximal set of commuting observables defined in terms of the operators

$$\vec{K}_r = \sum_{n=1}^{r} \vec{S}_n, \qquad r = 1, \ldots, N - 2, \tag{145}$$

which are the generator of $SU(2)$ rotations of the first $r$ spins.

The dimension of the Hilbert space $\mathcal{H}_G^{(j)}$ of the internal degrees of freedom can be computed using the general formula for the dimension of $SU(2)$ intertwiner space between the representations $j_1, \ldots, j_L$, (see App. A for a detailed derivation):

$$D_j \equiv \dim \mathcal{H}_G^{(j)} = \frac{2j+1}{j + \frac{N}{2} + 1} \binom{N}{\frac{N}{2} + j}, \tag{146}$$

where $\binom{n}{k} = \frac{n!}{k!(n-k)!}$ is the binomial coefficient. The sum of the dimensions of symmetry-resolved sectors matches the dimension $2^N$ of the Hilbert space of $N$ spin-$\frac{1}{2}$ particles, i.e., $\dim \mathcal{H}_N = \sum_j (2j+1) D_j = 2^N$.

*Symmetry-resolved states* are states of $N$ spin-$\frac{1}{2}$ particles that transform in a spin-$j$ representation of the total spin and have no entanglement between its rotational and its internal degrees of freedom, i.e., states of the form:

$$|\psi\rangle = |j, \xi\rangle_{\text{sym}} |j, \chi\rangle_G \in \mathcal{H}_N, \tag{147}$$

with

$$|j, \xi\rangle_{\text{sym}} = \sum_m \xi_m |j, m\rangle \in \mathcal{H}_{\text{sym}}^{(j)}, \tag{148}$$

$$|j, \chi\rangle_G = \sum_{k_1, \ldots, k_{N-2}} c_{k_1, \ldots, k_{N-2}} |j, k_1, \ldots, k_{N-2}\rangle \in \mathcal{H}_G^{(j)}, \tag{149}$$

i.e.,

$$|\psi\rangle = \sum_{m, k_1, \ldots, k_{N-2}} \xi_m \, c_{k_1, \ldots, k_{N-2}} |j, m\rangle |j, k_1, \ldots, k_{N-2}\rangle. \tag{150}$$

## 5.2 Symmetry-resolved subsystems: $K$-local vs $G$-local observables

A system of $N$-particles often comes with a built-in notion of locality. For instance, the particles might be distributed at the nodes of a lattice, therefore inducing a notion of first-neighbors and regions. Local observables of the system can then be expressed in terms of the spin operators $\vec{S}_n$ of a subset of spins belonging to the region. The kinematical algebra of observables $\mathcal{A}_K$ is generated by the complete set of spin operators (136), while the subalgebra of observables in a region $A$ is generated by the $N_A$ spin operators in the region,

$$\mathcal{A}_{KA} = \mathbb{C}[S_a^i], \qquad \text{with} \qquad a = 1, \ldots, N_A. \tag{151}$$

We call $\mathcal{A}_{KA}$ the *K-local* subalgebra of observables for the region $A$. Note that this subalgebra induces the standard decomposition of the Hilbert space as a tensor product $\mathcal{H}_N = \mathcal{H}_A \otimes \mathcal{H}_B$ over the region $A$ and its complement $B$. In fact one finds that the commutant of $\mathcal{A}_{KA}$ is generated by the observables in the complementary region $B$, with $N_A + N_B = N$ and

$$\mathcal{A}_{KB} \equiv (\mathcal{A}_{KA})' = \mathbb{C}[S_b^i], \qquad \text{with} \qquad b = N_A + 1, \ldots, N_A + N_B. \tag{152}$$

Moreover, note that the center of the subalgebra is trivial, $\mathcal{A}_{KA} \cap \mathcal{A}_{KB} = \mathbb{1}$, which results in the familiar tensor product structure with no direct sum.

Observables that are invariant under the group $G = SU(2)$ form the algebra $\mathcal{A}_G$, (138). To identify a local subalgebra of $G$-invariant observables we take the intersection with $\mathcal{A}_{KA}$,

$$\mathcal{A}_{GA} \equiv \mathcal{A}_G \cap \mathcal{A}_{KA} = \mathbb{C}[\vec{S}_a \cdot \vec{S}_{a'}], \qquad \text{with} \qquad a, a' = 1, \ldots, N_A. \tag{153}$$

We call $\mathcal{A}_{GA}$ the *G-local* subalgebra of observables for the region $A$. Note that this subalgebra is generated by the scalar products $\vec{S}_a \cdot \vec{S}_{a'}$ of spins in $A$. The commutant of $\mathcal{A}_{GA}$ in $\mathcal{A}_K$ is

$$\mathcal{A}_{\overline{GA}} \equiv (\mathcal{A}_{GA})' = (\mathcal{A}_G \cap \mathcal{A}_{KA})' = ((\mathcal{A}_{\text{sym}})' \cap (\mathcal{A}_{KB})')' \tag{154}$$

$$= (\mathbb{C}[J^i]' \cap \mathbb{C}[S_b^i]')' = \mathbb{C}[J^i, S_b^i] = \mathbb{C}[J^i, L^i, S_b^i], \tag{155}$$

where we have used the intersection formula (15). Note that $\mathcal{A}_{\overline{GA}}$ is not a subalgebra of $\mathcal{A}_{KB}$ as it contains also the total symmetry generator $J^i$. Note also that $\mathcal{A}_{\overline{GA}}$ is not the same as $\mathcal{A}_{GB}$. As we have highlighted in the last equality above, it is useful also to introduce the operator $L^i$ that measures the total spin of the particles in $A$,

$$\vec{L} = \sum_{a=1}^{N_A} \vec{S}_a, \tag{156}$$

and generates rotations of the spins in $A$. Note that $[\vec{L}^2, J^i] = 0$ because $J^i$ generates global rotations and $\vec{L}^2$ is rotationally invariant. Therefore, the center of the subalgebra is non-trivial

$$\mathcal{Z}_{GA} = \mathcal{A}_{GA} \cap \mathcal{A}_{\overline{GA}} = \mathbb{C}[\vec{L}^2], \tag{157}$$

and is generated by the Casimir operator $\vec{L}^2$ of $A$. We denote its eigenvalues $\ell(\ell+1)$ with

$$N_A \text{ even} \implies \ell_{\min} = 0, \qquad \ell_{\max} = \tfrac{N_A}{2}, \qquad \ell \text{ integer}, \tag{158}$$

$$N_A \text{ odd} \implies \ell_{\min} = \tfrac{1}{2}, \qquad \ell_{\max} = \tfrac{N_A}{2}, \qquad \ell \text{ half-integer}. \tag{159}$$

Using the results of Sec. 3 on subsystems from subalgebras, we find the decomposition

$$\mathcal{H}_N = \bigoplus_{\ell=\ell_{\min}}^{\ell_{\max}} \left( \mathcal{H}_{GA}^{(\ell)} \otimes \mathcal{H}_{\overline{GA}}^{(\ell)} \right). \tag{160}$$

This decomposition is useful for computing the reduction of a generic pure state of the system to the $G$−local subalgebra. We are interested in the reduction of a special class of states—symmetry-resolved states—and it is useful to introduce a decomposition adapted to them.

We prepare a symmetry-resolved state $|\psi\rangle = |j, \xi\rangle_{\text{sym}} |j, \chi\rangle_G$ and we are interested in measurements of $G$-local observables in $A$. In order to build a basis adapted to both the decomposition associated with $\mathcal{A}_{\text{sym}} = \mathbb{C}[J^i]$ and $\mathcal{A}_{GA} = \mathbb{C}[\vec{S}_a \cdot \vec{S}_{a'}]$, we consider the algebra generated by the union of the generating sets,

$$\mathcal{A}_{\text{sym}GA} = \mathbb{C}[J^i, \vec{S}_a \cdot \vec{S}_{a'}]. \tag{161}$$

Note that, using (15), this algebra can also be written in terms of the intersection of commutants,

$$\mathcal{A}_{\text{sym}GA} = \left( (\mathcal{A}_{\text{sym}})' \cap (\mathcal{A}_{GA})' \right)'. \tag{162}$$

To build the decomposition, we need its commutant and center. The commutant of $\mathcal{A}_{\text{sym}GA}$ in $\mathcal{A}_K$ is

$$(\mathcal{A}_{\text{sym}GA})' = (\mathcal{A}_{\text{sym}})' \cap (\mathcal{A}_{GA})' = \mathcal{A}_G \cap \mathcal{A}_{\overline{GA}} = \mathbb{C}[\vec{S}_n \cdot \vec{S}_{n'}] \cap \mathbb{C}[J^i, L^i, S_b^i] = \mathbb{C}[\vec{J}^2, \vec{L}^2, \vec{S}_b \cdot \vec{S}_{b'}]. \tag{163}$$

The center of $\mathcal{A}_{\text{sym}GA}$ is then non-trivial,

$$\mathcal{Z}_{\text{sym}GA} = \mathcal{A}_{\text{sym}GA} \cap (\mathcal{A}_{\text{sym}GA})' = \mathbb{C}[\vec{J}^2, \vec{L}^2]. \tag{164}$$

Note that the observables $\vec{J}^2$ and $\vec{L}^2$ commute,[9] which is always the case for the center as it is an Abelian algebra by definition. Simultaneously diagonalizing the observables $\vec{J}^2$ and $\vec{L}^2$, we find the decomposition of the Hilbert space as a sum over $j$ and $\ell$

$$\mathcal{H}_N = \bigoplus_{j=j_{\min}}^{j_{\max}} \bigoplus_{\ell=\ell_{\min}}^{\ell_{\max}} \left( \mathcal{H}_{\text{sym}}^{(j)} \otimes \mathcal{H}_{GA}^{(\ell)} \otimes \mathcal{H}_{GB}^{(j,\ell)} \right), \tag{165}$$

which can be reorganized as

$$\mathcal{H}_N = \bigoplus_{j=j_{\min}}^{j_{\max}} \mathcal{H}_N^{(j)}, \qquad \text{with} \qquad \mathcal{H}_N^{(j)} = \mathcal{H}_{\text{sym}}^{(j)} \otimes \bigoplus_{\ell=\ell_{\min}}^{\ell_{\max}} \left( \mathcal{H}_{GA}^{(\ell)} \otimes \mathcal{H}_{GB}^{(j,\ell)} \right), \tag{166}$$

which provides a derivation of the expression (8) in the introduction. The Hilbert spaces appearing in (166) are defined by

$$\mathcal{H}_{GA}^{(\ell)} = \text{Inv}_G \big( \mathcal{H}^{(\ell)} \otimes \underbrace{\mathcal{H}^{(1/2)} \otimes \cdots \otimes \mathcal{H}^{(1/2)}}_{N_A} \big), \tag{167}$$

$$\mathcal{H}_{GB}^{(j,\ell)} = \text{Inv}_G \big( \mathcal{H}^{(j)} \otimes \mathcal{H}^{(\ell)} \otimes \underbrace{\mathcal{H}^{(1/2)} \otimes \cdots \otimes \mathcal{H}^{(1/2)}}_{N_B} \big). \tag{168}$$

---

[9] The fact that the commutator $[\vec{J}^2, \vec{L}^2] = 0$ vanishes can be quickly shown by noticing that $[J^i, \vec{L}^2] = 0$ as $J^i$ generates rotations of the full system and $\vec{L}^2$ is a scalar.

Their dimensions can be computed using the general formula for the dimension of $SU(2)$ intertwiner space (see App. A for a detailed derivation):

$$d_\ell \equiv \dim \mathcal{H}_{GA}^{(\ell)} = \frac{2\ell+1}{\frac{N_A}{2}+\ell+1}\binom{N_A}{\frac{N_A}{2}+\ell}, \tag{169}$$

$$b_{j\ell} \equiv \dim \mathcal{H}_{GB}^{(j,\ell)} = \binom{N-N_A}{\frac{N-N_A}{2}+j-\ell} - \frac{\frac{N-N_A}{2}-j-\ell}{\frac{N-N_A}{2}+j+\ell+1}\binom{N-N_A}{\frac{N-N_A}{2}+j+\ell}. \tag{170}$$

If we compare the decomposition (141) with (166), we obtain the decomposition of the Hilbert space of $G$-invariant degrees of freedom at fixed total angular momentum,

$$\mathcal{H}_G^{(j)} = \bigoplus_{\ell=\ell_{\min}}^{\ell_{\max}} \left(\mathcal{H}_{GA}^{(\ell)} \otimes \mathcal{H}_{GB}^{(j,\ell)}\right), \tag{171}$$

with the adapted orthonormal basis [23]

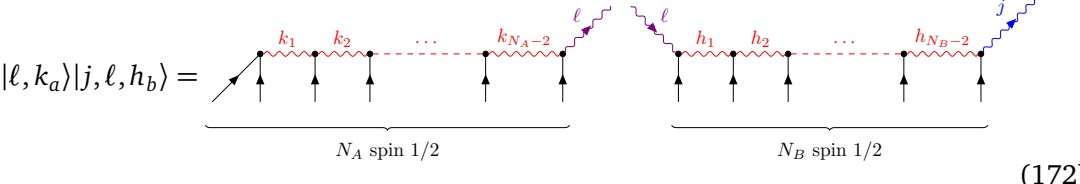

$$(172)$$

which coincides with (144), $|j, k_r\rangle$ with $k_r$ equal to $k_a$ for $r = 1, \ldots, N_A - 2$, equal to $\ell$ for $r = N_A - 1$, and equal to $h_b$ for $r = N_A, \ldots, N_A + N_B - 2$. From this decomposition, we also derive the relation between the dimensions $d_\ell$, $b_{j\ell}$ and $D_j$,

$$D_j = \sum_\ell d_\ell \, b_{j\ell}. \tag{173}$$

We can now compute the density matrix $\rho_{GA}$ of a symmetry-resolved state $|\psi\rangle$ reduced to $\mathcal{A}_{GA}$. Following the general construction described in Sec. 3, the generic symmetry-resolved state in this basis is:

$$|\psi\rangle = |j,\xi\rangle \sum_\ell \sqrt{p_\ell} \sum_{k_a,h_b} \chi_{k_a,h_b}^{(\ell)} |\ell,k_a\rangle|j,\ell,h_b\rangle. \tag{174}$$

The density matrix of the symmetry-resolved state reduced to $\mathcal{A}_{GA}$ is

$$\rho_{GA} = \bigoplus_\ell p_\ell \, \rho_\ell, \tag{175}$$

where

$$\rho_\ell = \sum_{k_a,k_a'} \sum_{h_b} \chi_{k_a,h_b}^{(\ell)} \chi_{k_a',h_b}^{(\ell)*} |\ell,k_a\rangle\langle\ell,k_a'|. \tag{176}$$

This is the expression used in the next section to compute the $G$-local entanglement entropy $S_{GA}$.

We conclude this section with a few observations. By direct comparison of the decomposition (160) and (165), we see that the decomposition of the complement of the $GA$-Hilbert space is not the $GB$-Hilbert space as

$$\mathcal{H}_{\overline{GA}}^{(\ell)} = \bigoplus_j \left(\mathcal{H}_{\text{sym}}^{(j)} \otimes \mathcal{H}_{GB}^{(j,\ell)}\right). \tag{177}$$

Moreover, differently from what happens in the Abelian case discussed in Sec. 4.3, we note that here states in $\mathcal{H}_{GB}^{(j,\ell)}$ are eigenstates of the total angular momentum $\vec{J}^2$ and of the angular momentum of $A$, i.e., $\vec{L}^2$, but they are not in an eigenstate of the angular momentum of $B$, i.e., $\vec{J}_B^2 = (\sum_b \vec{S}_b)^2$, unless $\vec{J}^2 = 0$ in which case $\vec{J}_B^2 = \vec{L}^2$.

### 5.3 $G$-local entanglement entropy of symmetry-resolved states

Given a symmetry-resolved state, i.e., a state of the form (174), we can compute its density matrix reduced to the $G$-local subalgebra of observables $\mathcal{A}_{GA}$ using the techniques of the previous section. The $G$-local entanglement entropy is then given by the general formula (29),

$$S_{GA} \equiv S\big(|\psi\rangle, \mathcal{A}_G, \mathcal{A}_{GA}\big) = -\operatorname{tr}(\rho_{GA} \log \rho_{GA}) \tag{178}$$

$$= -\sum_\ell p_\ell \operatorname{tr}(\rho_\ell \log \rho_\ell) - \sum_\ell p_\ell \log p_\ell \,, \tag{179}$$

where $\rho_\ell$ is the reduced density matrix in the sector $\ell$ and $p_\ell$ the probability of the sector, as defined in (175). We give a few examples to illustrate how $S_{GA}$ is computed and its differences from $S_{KA}$:

$N = 2$, $j = 0$ — This is the singled state $|s\rangle$ of two spin-1/2 particles,

$$|s\rangle = \frac{|\uparrow\downarrow\rangle - |\downarrow\uparrow\rangle}{\sqrt{2}}\,. \tag{180}$$

As $K$-local subsystem with $N_A = 1$, we can take the first particle with the algebra of observables $\mathcal{A}_{KA} = \mathbb{C}[\vec{S}_1]$. As usual [16], the associated entanglement entropy is $S_{KA}(|s\rangle) = \log 2$. On the other hand, if we consider the algebra of $G$-local observables of the first particle we find that it is trivial as there is no rotational invariant observable besides $\vec{S}_1^2 = \frac{1}{2}(\frac{1}{2} + 1)\mathbb{1}$, i.e., $\mathcal{A}_{GA} = \{\mathbb{1}\}$ and $S_{GA}(|s\rangle) = 0$.

$N = 2$, $j = 1$ — This is the Hilbert space of the triplet state $|t_m\rangle$,

$$|t_m\rangle = \begin{cases} |\uparrow\uparrow\rangle, & m = +1, \\ \frac{|\uparrow\downarrow\rangle + |\downarrow\uparrow\rangle}{\sqrt{2}}, & m = 0, \\ |\downarrow\downarrow\rangle, & m = -1. \end{cases} \tag{181}$$

The $K$-local subsystem with $N_A = 1$ has entanglement entropy $S_{KA}(|t_\pm\rangle) = 0$ and $S_{KA}(|t_0\rangle) = \log 2$. On the other hand, the $G$-local entanglement entropy $S_{GA}(|t_m\rangle) = 0$ vanishes again as the subalgebra $\mathcal{A}_{GA} = \{\mathbb{1}\}$ is trivial.

$N = 4$, $j = 0$ — The recoupling of $N = 4$ spin-1/2 particles into a scalar ($j = 0$) defines a two-dimensional Hilbert space spanned by the two orthogonal states

$$|\psi_0\rangle = |s\rangle_A |s\rangle_B\,, \tag{182}$$

$$|\psi_1\rangle = \sum_{m=0,\pm 1} \frac{(-1)^{1-m}}{\sqrt{3}} |t_{+m}\rangle_A |t_{-m}\rangle_B\,, \tag{183}$$

where we have denoted $A = (1, 2)$ and $B = (3, 4)$ the coupling of the four particles. The $K$-local and the $G$-local entanglement entropies of the subsystem $A$ are

$$\begin{aligned} S_{KA}(|\psi_0\rangle) &= 0\,, & S_{GA}(|\psi_0\rangle) &= 0\,, \\ S_{KA}(|\psi_1\rangle) &= \log 3\,, & S_{GA}(|\psi_1\rangle) &= 0\,. \end{aligned} \tag{184}$$

Note that the $K$-local entropy measures the entanglement in the magnetic degrees of freedom $m$, which results in the $\log 3$ above. On the other hand, the $G$-local entanglement entropy for each of the two states vanishes as they are eigenstates of $(\vec{S}_1 + \vec{S}_2)^2$, which is the only non-trivial observable in $\mathcal{A}_{GA} = \mathbb{C}[\vec{S}_1 \cdot \vec{S}_2]$. In fact, using the basis (172), we see that $|\psi_1\rangle$ is factorized. To show a non-trivial $G$-local entropy, we consider the superposition

$$|\psi\rangle = \sqrt{1-p}\,|\psi_0\rangle + \sqrt{p}\,\mathrm{e}^{\mathrm{i}\phi}\,|\psi_1\rangle\,, \tag{185}$$

for which we can easily compute the reduced density matrices

$$\rho_{KA} = (1-p)|s\rangle\langle s| + p \sum_{m=0,\pm 1} \tfrac{1}{3}|t_m\rangle\langle t_m|, \tag{186}$$

$$\rho_{GA} = (1-p)|0\rangle\langle 0| + p|1\rangle\langle 1|, \tag{187}$$

where the states $|0\rangle$ and $|1\rangle$ are the eigenstates with $\ell = 0,1$ of the $G$-local observable $\vec{L}^2 = (\vec{S}_1 + \vec{S}_2)^2$. It follows that the entanglement entropies for the $K$-local and the $G$-local subalgebras are

$$S_{KA}(|\psi\rangle) = -(1-p)\log(1-p) - p\log(p/3), \tag{188}$$

$$S_{GA}(|\psi\rangle) = -(1-p)\log(1-p) - p\log p. \tag{189}$$

In this simple case, the $G$-local entanglement entropy is purely due to the Shannon entropy of the sector, i.e., the last term in (179), because the subalgebra coincides with the center, $\mathcal{A}_{GA} = \mathcal{A}_{GB} = \mathcal{Z}$.

$N = 4, j = 1, m = +1$ — The recoupling of $N = 4$ spin-1/2 particles into a vector ($j = 1$) defines the Hilbert space $\mathcal{H}_4^{(1)} = \mathcal{H}_{\text{sym}}^{(1)} \otimes \mathcal{H}_G^{(1)}$ of dimension $3 \times 3$. We consider a ($m = +1$) symmetry-resolved state

$$|\psi\rangle = \sqrt{1-p}\,|\eta_1\rangle + \sqrt{p}\,e^{i\phi}\,|\eta_2\rangle, \tag{190}$$

given by the superposition of the two orthogonal basis states with $m = +1$

$$|\eta_1\rangle = |s\rangle_A |t_+\rangle_B, \qquad |\eta_2\rangle = \tfrac{1}{\sqrt{2}}\left(|t_+\rangle_A |t_0\rangle_B - |t_0\rangle_A |t_+\rangle_B\right), \tag{191}$$

where $|s\rangle$ and $|t_m\rangle$ are the singlet state and the triplet state (with the magnetic number $m = 0,\pm 1$) obtained by coupling two spin-1/2 particles. The $K$-local density matrix for the two spins in $A$, defined as usual as $\rho_{KA} = \text{Tr}_B |\psi\rangle\langle\psi|$, is

$$\rho_{KA} = (1-p)|s\rangle\langle s| + \tfrac{p}{2}\left(|t_+\rangle\langle t_+| + |t_0\rangle\langle t_0|\right) - \sqrt{\tfrac{p(1-p)}{2}}\left(e^{-i\phi}|s\rangle\langle t_0| + e^{+i\phi}|t_0\rangle\langle s|\right). \tag{192}$$

The non-vanishing eigenvalues of $\rho_{KA}$ are $\{\tfrac{p}{2}, 1-\tfrac{p}{2}\}$. Therefore, $K$-local entanglement entropy is

$$S_{KA}(|\psi\rangle) = -\tfrac{p}{2}\log\tfrac{p}{2} - (1-\tfrac{p}{2})\log(1-\tfrac{p}{2}). \tag{193}$$

On the other hand, if we have only access to the rotational invariant observable of the particles in subsystem $A = (1,2)$, that is only the observable $\vec{S}_1 \cdot \vec{S}_2$, then the accessible entropy is given by the probability of outcomes $\{p, 1-p\}$,

$$S_{GA}(|\psi\rangle) = -p\log p - (1-p)\log(1-p). \tag{194}$$

We note that, for $\tfrac{2}{3} \le p \le 1$, we have that the $G$-local entropy is smaller than the $K$-local entropy, $S_{GA}(|\psi\rangle) \le S_{KA}(|\psi\rangle)$, while for $0 \le p \le \tfrac{2}{3}$ the $G$-local entropy is larger $S_{GA}(|\psi\rangle) \ge S_{KA}(|\psi\rangle)$. We note also another difference compared to the Abelian case (1) where the reduced density matrix is shown to be black diagonal. From (192), we see that the density matrix $\rho_{KA}$ is not block diagonal as it has non-vanishing matrix elements $|s\rangle\langle t_0|$ that connect blocks with different $\ell$. This is a generic feature of $SU(2)$-symmetry-resolved states.

$N = 6, j = 1, m = +1$ — As a last example, we illustrate the case considered in the introduction in Fig. 3. The Hilbert space $\mathcal{H}_6^{(1)} = \mathcal{H}_{\text{sym}}^{(1)} \otimes \mathcal{H}_G^{(1)}$ has dimension $3 \times 9$. We consider the subsystem of the first $N_A = 3$ particles. The $G$-local entanglement entropy is associated to the restriction of the state to the subalgebra $\mathcal{A}_{GA} = \mathbb{C}[\vec{S}_1 \cdot \vec{S}_2, \vec{S}_1 \cdot \vec{S}_3, \vec{S}_2 \cdot \vec{S}_3]$ which now is a

non-commutative algebra and therefore allows intertwiner entanglement, i.e., the first term in (179). The associated Hilbert space decomposition is

$$\mathcal{H}_G^{(1)} = \bigoplus_{\ell=\frac{1}{2},\frac{3}{2}} \left( \mathcal{H}_{GA}^{(\ell)} \otimes \mathcal{H}_{GB}^{(1,\ell)} \right). \tag{195}$$

We consider the symmetry-resolved state with $m = +1$ described in Fig. 3(b),

$$|\psi\rangle = \tfrac{1}{\sqrt{2}}(|\psi_1\rangle + |\psi_2\rangle), \tag{196}$$

given by the superposition of the two orthonormal $|\psi_1\rangle$ and $|\psi_2\rangle$ introduced in (11). As shown in Fig. 3(b) for $p = 1/2$, the $G$-local entropy of this state is $S_{GA}(|\psi\rangle) = \log 2$ while the $K$-local entropy is $S_{KA}(|\psi\rangle) = -\frac{3}{8}\log\frac{3}{8} - \frac{5}{8}\log\frac{5}{8}$. This example allows us to comment on the symmetry under the exchange of $A$ with $B$. Clearly $S_{KA}(|\psi\rangle) = S_{KB}(|\psi\rangle)$. On the other hand, we note that the restriction to the subalgebra $\mathcal{A}_{GB} = \mathbb{C}[\vec{S}_4 \cdot \vec{S}_5, \vec{S}_4 \cdot \vec{S}_6, \vec{S}_5 \cdot \vec{S}_6]$ is associated with Hilbert space decomposition

$$\mathcal{H}_G^{(1)} = \bigoplus_{j_B=\frac{1}{2},\frac{3}{2}} \left( \mathcal{H}_{GB}^{(j_B)} \otimes \mathcal{H}_{GA}^{(1,j_B)} \right), \tag{197}$$

and $S_{GB}(|\psi\rangle) = 0$ because $|\psi\rangle$ is an eigenstate of $(\vec{S}_5 + \vec{S}_6)^2 = 0$ and $(\vec{S}_4 + \vec{S}_5 + \vec{S}_6)^2 = \frac{1}{2}(\frac{1}{2}+1)$. This example shows concretely that, in the non-Abelian case, the commutant symmetry (39) allows an asymmetry under subsystem exchange in the $G$-local entanglement entropy, $S_{GB}(|\psi\rangle) \neq S_{GA}(|\psi\rangle)$.

## 5.4 Typical $G$-local entanglement entropy: Exact formulas

We combine the results of Sec. 2.3 on the typical entanglement entropy of a subsystem defined in terms of a subalgebra of observables, together with the results of Sec. 3.3 on symmetry-resolved entanglement entropy, to derive the exact formulas for the typical $G$-local entanglement entropy for the group $G = SU(2)$.

The average entanglement entropy for a random symmetry-resolved state in $\mathcal{H}_N^{(j)} = \mathcal{H}_{\text{sym}}^{(j)} \otimes \mathcal{H}_G^{(j)}$, restricted to a $G$-local subsystem of $N_A$ spin-1/2 particles is given by the formula (43), specialized to the $SU(2)$ case:

$$\langle S_{GA} \rangle_j = \sum_{\ell=\ell_{\min}}^{\ell_{\max}} \varrho_\ell \, \varphi_\ell, \tag{198}$$

where $\varrho_\ell$ and $\varphi_\ell$ are given in terms of the dimensions $d_\ell, b_{j\ell}, D_j$ defined in (146), (169), (170). The first quantity is

$$\varrho_\ell = \frac{d_\ell \, b_{j\ell}}{D_j}, \tag{199}$$

and the second is

$$\varphi_\ell = \Psi(D_j+1) - \Psi(\max(d_\ell, b_{j\ell})+1) - \min\left(\frac{d_\ell-1}{2b_{j\ell}}, \frac{b_{j\ell}-1}{2d_\ell}\right). \tag{200}$$

The variance $(\Delta S_{GA})^2 = \langle S_{GA}^2 \rangle_j - \langle S_{GA} \rangle_j^2$ can be written as

$$(\Delta S_{GA})^2 = \frac{1}{D+1}\left( \sum_{\ell=\ell_{\min}}^{\ell_{\max}} \varrho_\ell \left( \varphi_\ell^2 + \chi_\ell \right) - \left( \sum_{\ell=\ell_{\min}}^{\ell_{\max}} \varrho_\ell \, \varphi_\ell \right)^2 \right), \tag{201}$$

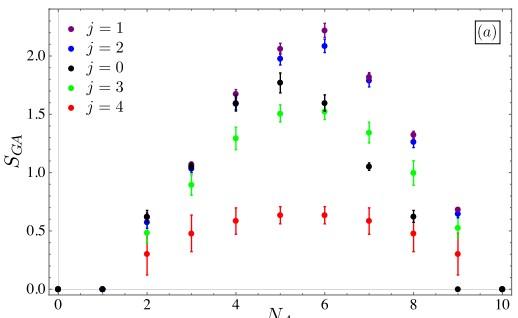 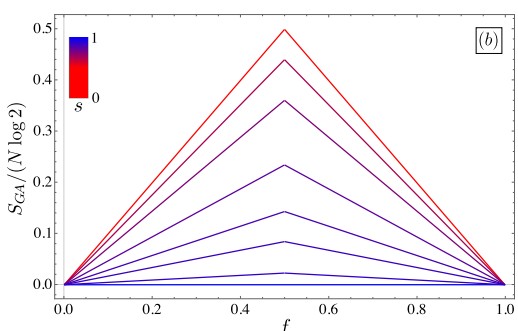

Figure 4: (a) Page curve for the $SU(2)$ symmetry-resolved entanglement entropy in a system consisting of $N = 10$ spins. The average entanglement entropy $\langle S_{GA} \rangle_j$ and its dispersion $(\Delta S_{GA})_j$ are computed using the exact formulas (198)–(201) and reported as a function of the number of spins $N_A$ in the subsystem. The Page curve with $j = 1$ has the largest peak entropy. The curves with $j = 2$, $j = 0$, $j = 3$ and $j = 4$ follow. The maximum spin $j = 5$ has $S_{GA} = 0$. The ordering of the curves reflects the dimension of the Hilbert spaces $D_j$. For $N_A = 1$, the $G$-local subsystem is trivial, $d_\ell = 1$, and the entropy $S_{GA} = 0$ vanishes. The Page curve is generally not symmetric under the exchange $N_A \leftrightarrow N - N_A$, except in the special case $j = 0$ where this exchange symmetry is present. (b) Leading order of the symmetry-resolved entanglement entropy in the thermodynamic limit. At this order, the entanglement entropy is symmetric under exchange $f \leftrightarrow 1 - f$. We plot the Page curve for spin densities $s = 0$, $s = 0.4$, $s = 0.6$, $s = 0.8$, $s = 0.9$, $s = 0.95$, $s = 0.99$, and $s = 1$, which corresponds to curves from the top to the bottom.

where $\varrho_\ell$ and $\varphi_\ell$ are given above and $\chi_\ell$ is defined as

$$
\begin{aligned}
\chi_\ell =\ & (d_\ell + b_{j\ell}) \Psi'(\max(d_\ell, b_{j\ell}) + 1) - (D_j + 1) \Psi'(D_j + 1) \\
& - \frac{(\min(d_\ell, b_{j\ell}) - 1)(d_\ell + b_{j\ell} + \max(d_\ell, b_{j\ell}) - 1)}{4 \max(d_\ell^2, b_{j\ell}^2)}.
\end{aligned}
\tag{202}
$$

The formulas for average (198) and the variance (201) of the symmetry-resolved entanglement entropy of a random state are exact and can be computed from the expressions of the dimensions of the Hilbert spaces $d_\ell$, $b_{j\ell}$, and $D_j$.

In Fig. 3(a), we show the average and variance compared to the statistical distribution of a sample of symmetry-resolved random states with $N = 6$, $j = 1$, $m = +1$ restricted to $N_A = 3$.

In Fig. 4(a), we show the exact average and variance for $N = 10$ and different values of $j$ as a function of the subsystem size $N_A$, i.e., the Page curve [1, 2] for a symmetry-resolved system. Note the asymmetry under exchange $N_A \rightarrow N - N_A$, due to the asymmetry in the dimensions $d_\ell$ and $b_{j\ell}$ in (169)–(170), which is a generic feature of non-Abelian symmetry-resolved entanglement (see Sec. 4.2).

As discussed in (100)–(104), one can also decompose the total symmetry-resolved entanglement entropy $\langle S_{GA} \rangle_j$ in a configurational $\langle S_{GA}^{(conf)} \rangle_j$ and a number contribution $\langle S_{GA}^{(num)} \rangle_j$.

## 6 Large $N$ asymptotics of the $SU(2)$ typical entropy

In a lattice many-body system, it is useful to introduce intensive quantities that allow us to take a thermodynamic limit $N \rightarrow \infty$ and study the behavior of the subsystem entropy as a function of the fraction of the lattice. In this section we study the $SU(2)$ symmetry-resolved entanglement entropy in this limit.

## 6.1 Fixed spin density

The symmetry-resolved entanglement entropy (198) depends on the system size $N$, the total spin $j$, and the subsystem size $N_A$. Moreover, the expression contains a sum over the subsystem spin $\ell$. We introduce intensive quantities representing the subsystem fraction $f$, the system spin density $s$, and the subsystem spin density $t$:

$$f = \frac{N_A}{N}, \qquad s = \frac{2j}{N}, \qquad t = \frac{2\ell}{N_A}. \tag{203}$$

The thermodynamic limit is defined as the limit $N \to \infty$ while keeping the intensive quantities $f$ and $s$ fixed. We compute the average entropy (198) and its variance (201) in this limit up to terms of order one, $O(1)$.

First, we derive the asymptotic expressions of the dimensions $d_\ell$, $b_{j\ell}$, and $D_j$ in the thermodynamic limit. We use the asymptotic expansion of the binomial coefficients for $n \to \infty$ with $\lambda$ fixed,

$$\binom{n}{\frac{n}{2}(1+\lambda)} = \sqrt{\frac{2 \, |\beta''(\lambda)|}{\pi n}} \, e^{n\beta(\lambda)} \left(1 - \frac{1}{12n} \frac{3 + \lambda^2}{1 - \lambda^2} + O(n^{-2})\right), \tag{204}$$

where the function $\beta(s)$ is defined by

$$\beta(s) = -\frac{1-s}{2} \log\left(\frac{1-s}{2}\right) - \frac{1+s}{2} \log\left(\frac{1+s}{2}\right), \tag{205}$$

and its derivatives are

$$\beta'(s) = -\frac{1}{2} \log\left(\frac{1+s}{1-s}\right), \qquad \beta''(s) = -\frac{1}{1-s^2}. \tag{206}$$

The approximation (204) holds for $\lambda = O(1)$ and is useful for deriving the large-$N$ asymptotics of the dimensions at fixed $0 < s < 1$. This approximation is invalid in the extremal cases $j = j_{\min}$ and $j = j_{\max}$ defined in (134), and we will deal with them separately in the next section. The asymptotic form of the dimensions in terms of intensive quantities are

$$D_j = \frac{2s}{1+s} \sqrt{\frac{2 \, |\beta''(s)|}{\pi N}} \, e^{N\beta(s)} \left(1 + \frac{1}{12N} \frac{12 - s(9-s)(3-s)}{s(1-s^2)} + O(N^{-2})\right), \tag{207}$$

$$d_\ell = \frac{2t}{1+t} \sqrt{\frac{2 \, |\beta''(t)|}{\pi f N}} \, e^{f N \beta(t)} \left(1 + \frac{1}{12 f N} \frac{12 - t(9-t)(3-t)}{t(1-t^2)} + O(N^{-2})\right), \tag{208}$$

$$b_{j\ell} = \sqrt{\frac{2 \, |\beta''(\frac{s-tf}{1-f})|}{\pi N (1-f)}} \, e^{(1-f)N\beta(\frac{s-tf}{1-f})} \left(1 - \frac{1}{12(1-f)N} \frac{3 + \left(\frac{s-tf}{1-f}\right)^2}{1 - \left(\frac{s-tf}{1-f}\right)^2} + O(N^{-2})\right). \tag{209}$$

Note that, in the asymptotic formula for $b_{j\ell}$, the second term in the exact expression (170) is exponentially small, and therefore it does not contribute to the asymptotics for $0 < s < 1$ and large $N$. On the other hand, this approximation is invalid for $s = 0$, that is, when $j = j_{\min}$ which is not compatible with the scaling assumed in (203), and will be treated separately in the next section. In the thermodynamic limit, the sum over the spin $\ell$ becomes an integral over the spin density $\sum_\ell \to \int_0^1 f \frac{N}{2} dt$ and the probability distribution (199) is given by a continuous probability distribution

$$\varrho(t)dt = \frac{d_\ell b_{j\ell}}{D_j} f \frac{N}{2} dt = \varrho_0(t)\left(1 + \frac{\varrho_1(t)}{N} + O(N^{-2})\right) e^{N\left(f\beta(t) + (1-f)\beta\left(\frac{s-ft}{1-f}\right) - \beta(s)\right)} dt, \tag{210}$$

where

$$\varrho_0(t) = f\frac{N}{2}\sqrt{\frac{2}{\pi f(1-f)N}}\frac{(s+1)t}{s(t+1)}\sqrt{\frac{|\beta''(t)|\,|\beta''(\frac{s-ft}{1-f})|}{|\beta''(s)|}}, \tag{211}$$

$$\varrho_1(t) = \frac{1}{12f}\frac{12-t(9-t)(3-t)}{t(1-t^2)} - \frac{1}{12(1-f)}\frac{3+\left(\frac{s-tf}{1-f}\right)^2}{1-\left(\frac{s-tf}{1-f}\right)^2} - \frac{1}{12}\frac{12-s(9-s)(3-s)}{s(1-s^2)}. \tag{212}$$

The limit of function $\varphi_\ell$ (200) is given by the lower bound (50),

$$\varphi(t) = \log\min\left(\frac{D_j}{b_{j\ell}},\frac{D_j}{d_\ell}\right) - \frac{1}{2}\min\left(\frac{d_\ell}{b_{j\ell}},\frac{b_{j\ell}}{d_j}\right), \tag{213}$$

and the average entanglement entropy at fixed $s$ is given by the integral

$$\langle S_{GA}\rangle_s = \int_0^1 \varrho(t)\varphi(t)\,dt. \tag{214}$$

We evaluate the integral over $t$ using the Laplace approximation for large $N$ (see App. B). The integral is concentrated at the critical point $t = s$ defined as the maximum of the exponent of (210). If $f < \frac{1}{2}$, at the critical point the dimension $b_{j\ell}$ is exponentially larger than $d_j$. Therefore, we can ignore the second term in (213) as it is exponentially small. Similarly, at the critical point, the min in the logarithm in (213) selects the ratio $D_j/b_{j\ell}$, which allows us to write

$$\varphi(t) = N\left(\beta(s) - (1-f)\beta\left(\frac{s-ft}{1-f}\right)\right) + \frac{1}{2}\log\left(\frac{\beta''(s)}{\beta''\left(\frac{s-ft}{1-f}\right)}\right) + \log\left(\frac{2s\sqrt{1-f}}{1+s}\right) + O(N^{-1}). \tag{215}$$

At half-system size, $f = 1/2$, extra care is necessary because of a discontinuity in the integral. The dimensions satisfy the inequalities $b_{j\ell} > d_\ell$ for $t < s$, but $b_{j\ell} < d_\ell$ for $t > s$. The logarithm in (213) is discontinuous at the critical point $t = s$, and we have to resort to a Laplace approximation that allows for discontinuities (see App. B). There are additional contributions at order $O(\sqrt{N})$ and at order $O(1)$. Note that, at $f = 1/2$, the second term in (213) is not exponentially small, but detailed analysis shows that it is of order $O(1/\sqrt{N})$ and therefore does not contribute to the thermodynamic limit. Summarizing, the average entanglement entropy for $f \leq 1/2$ is:

$$\langle S_{GA}\rangle_s = \beta(s)fN - \frac{|\beta'(s)|}{\sqrt{2\pi|\beta''(s)|}}\sqrt{N}\,\delta_{f,\frac{1}{2}} + \frac{f+\log(1-f)}{2} + \left(1 - \frac{1}{2}\delta_{f,\frac{1}{2}}\right)\log\left(\frac{2s}{1+s}\right)$$

$$- \left(1 - f - \frac{1}{2}\delta_{f,\frac{1}{2}}\right)\frac{1-s}{2s}\log\left(\frac{1+s}{1-s}\right) + O(N^{-\frac{1}{2}}), \qquad \text{for} \quad f \leq \frac{1}{2}, \tag{216}$$

and for $f > 1/2$

$$\langle S_{GA}\rangle_s = \beta(s)(1-f)N + \frac{(1-f)+\log f}{2} + (1-f)\frac{1-s}{2s}\log\left(\frac{1+s}{1-s}\right) + O(N^{-\frac{1}{2}}), \qquad \text{for} \quad f > \frac{1}{2}. \tag{217}$$

Note that the leading order $O(N)$ and the subleading order $O(\sqrt{N})$ are symmetric under exchange of the subsystem with its complement. However, the order $O(1)$ term is not symmetric, $f \not\leftrightarrow (1-f)$. The leading order and its dependence on the spin density $s$ via the function $\beta(s)$ (205) is displayed in Fig. 4(b).

Using the same technique, we find that the variance is:

$$(\Delta S_{GA})_s^2 = \sqrt{\frac{\pi}{2}}\left(f(1-f) - \frac{1}{2\pi}\delta_{f,\frac{1}{2}}\right)\frac{(1-s)^{3/2}(s+1)^{5/2}}{8s}\left(\log\frac{1+s}{1-s}\right)^2 N^{\frac{3}{2}}e^{-N\beta(s)}\left(1 + O(N^{-1})\right). \tag{218}$$

We note that the variance is exponentially small in $N$. Moreover, at the order considered, the variance is invariant under the symmetry $f \leftrightarrow (1-f)$.

## 6.2   Extremal cases: $j_{\max}$ and $j_{\min}$

In the extremal case of maximum spin, $j = j_{\max} = N/2$, the Hilbert space $\mathcal{H}_G^{(j_{\max})}$ contains only one state and the dimension (146) is $D_{j_{\max}} = 1$. Moreover, the composition of angular momenta constrains the spin of the subsystem $A$ to be maximal, i.e., $\ell = \ell_{\max}$, and the dimensions (169) and (170) of the factors are $d_{\ell_{\max}} = 1$, $b_{j_{\max}\ell_{\max}} = 1$. Therefore, the unique symmetry-resolved state with $j = j_{\max}$ is factorized, as the exact formula of the average and variance of the entanglement entropy confirm:

$$\langle S_{GA}\rangle_{j_{\max}} = 0 \,, \qquad (\Delta S_{GA})^2_{j_{\max}} = 0 \,. \tag{219}$$

The other extremal case, $j = j_{\min}$, is non trivial. Let us assume that $N$ is even so that $j_{\min} = 0$. In this case, the relevant asymptotic formula for the binomial coefficient is

$$\binom{n}{\frac{n}{2}+x} = \sqrt{\frac{2}{n\pi}} \, \mathrm{e}^{n\log 2} \, \mathrm{e}^{-\frac{2x^2}{n}} \left(1 + O(n^{-1})\right), \tag{220}$$

that is valid for $x = O(\sqrt{n})$ and $n \to \infty$.[10] In the sum (198), we first assume $\ell = O(\sqrt{N})$, motivated by the fact that it corresponds to the subsystem with the largest dimension. We then check this hypothesis a posteriori. In this limit, the dimensions of the Hilbert spaces are

$$D_0 = \sqrt{\frac{8}{\pi}} \frac{1}{N^{3/2}} \, \mathrm{e}^{N\log 2} \left(1 + O(N^{-1})\right), \tag{221}$$

$$d_\ell = \sqrt{\frac{2}{\pi}} \frac{4\ell}{(fN)^{3/2}} \, \mathrm{e}^{fN\log 2} \, \mathrm{e}^{-2\frac{\ell^2}{fN}} \left(1 + O(N^{-1})\right). \tag{222}$$

Note that, for $j = 0$, the dimensions $b_{0\ell}$ and $d_\ell$ are mapped into each other by sending $f \leftrightarrow 1-f$. This is an exact property of (169) and (170) which in the asymptotic limit results into the expression

$$b_{0\ell} = \sqrt{\frac{2}{\pi}} \frac{4\ell}{((1-f)N)^{3/2}} \, \mathrm{e}^{(1-f)N\log 2} \, \mathrm{e}^{-2\frac{\ell^2}{(1-f)N}} \left(1 + O(N^{-1})\right). \tag{223}$$

It is useful to introduce the rescaled variable

$$u = \sqrt{\frac{2}{f(1-f)N}} \, \ell \,, \tag{224}$$

which simplifies the calculation of the integral. The probability distribution (199) in the thermodynamic limit becomes the continuous distribution

$$\varrho(u)\,\mathrm{d}u = \frac{4}{\sqrt{\pi}} u^2 \, \mathrm{e}^{-u^2} \left(1 + O(N^{-1})\right) \mathrm{d}u \,. \tag{225}$$

The sum over the spin $\ell$ becomes an integral over the positive real line in the thermodynamic limit. It is worth noticing that (225) is independent of the system's parameters $N$ and $f$ at the leading order and is normalized at all orders. The calculation of the average entropy is straightforward. If $f < \frac{1}{2}$, the dimension $b_{0\ell}$ is exponentially larger than $d_\ell$, therefore (213) is just given by the logarithmic term. If $f = \frac{1}{2}$, then $b_{0\ell} = d_\ell$ and the second term in (213) contributes a $-\frac{1}{2}$ correction. Therefore,

$$\varphi(u) = fN\log 2 - \frac{1}{2}\log N - \frac{1}{2}\log f - \frac{1}{2}\log 2 + \log(1-f) + u^2 f - \log u - \frac{1}{2}\delta_{f,\frac{1}{2}} + O(N^{-1}). \tag{226}$$

---

[10]Note that here we cannot use (204) where $\lambda = O(1)$.

We obtain the average entanglement entropy by performing the integral

$$\langle S_{GA} \rangle_0 = \int_0^\infty \varrho(u)\varphi(u)\,\mathrm{d}u, \tag{227}$$

keeping all terms up to $O(1)$. Summarizing, the average entanglement entropy for $f \leq \frac{1}{2}$ is

$$\begin{aligned}
\langle S_{GA} \rangle_0 =\ & fN \log 2 - \tfrac{1}{2} \log N - \tfrac{1}{2} \log f + \log(1-f) \\
& + (\tfrac{1}{2}-f)\log 2 + \tfrac{3}{2}f - 1 - \tfrac{\gamma_E}{2} - \tfrac{1}{2}\delta_{f,\frac{1}{2}} + O(N^{-1}).
\end{aligned} \tag{228}$$

The average entropy for $f > 1/2$ can be obtained via the exact symmetry $f \leftrightarrow (1-f)$ that applies to the case of $j = 0$. The calculation for the variance (201) is similar, and we find

$$(\Delta S_{GA})_0^2 = \tfrac{\sqrt{2\pi}}{4}\Big(f(\tfrac{3}{2}f - 1) + \tfrac{\pi^2}{8} - 1 + \tfrac{1}{4}\delta_{f,\frac{1}{2}}\Big)N^{\frac{3}{2}}\,e^{-N\log 2}\big(1 + O(N^{-1})\big), \tag{229}$$

for $f \leq 1/2$, and the formula for $f > 1/2$ can again be obtained via the exact symmetry $f \leftrightarrow (1-f)$ that applies to the case of $j = 0$.

Furthermore, we comment on the equipartition (or lack of equipartition) of entanglement entropy in the thermodynamic limit for the non-Abelian symmetry $SU(2)$, generalizing the discussion in [15] and [6] for the $U(1)$ case. By equipartition of entanglement, one means that the entropy $\langle S_{GA}^{(\ell)} \rangle_j$ is independent of $\ell$ in some limit. For generic spin $j$ (Sec. 6.1), we found in (215) that at the leading order at fixed subsystem charge $\ell$, the entanglement entropy is $\langle S_{GA}^{(\ell)} \rangle_j \approx N\big(\beta(s) - (1-f)\beta\big(\tfrac{s-ft}{1-f}\big)\big)$ with $s = 2j/N$ and $t = 2\ell/N_A$. Therefore, we conclude that there is no equipartition of entanglement entropy, as the leading order in $N$ depends explicitly on the subsystem charge $\ell$. This result extends the observation of [6] of lack of equipartition in the thermodynamic limit to the non-Abelian case. Furthermore, following the argument in [6], we emphasize the importance of the order of limits. If $\ell$ is fixed before taking the limit $N \to \infty$, using the expansion $\beta\big(\tfrac{s-ft}{1-f}\big) = \beta\big(\tfrac{s}{1-f}\big) - \beta'\big(\tfrac{s}{1-f}\big)\tfrac{ft}{1-f} + O(t^2)$, we obtain instead $\langle S_{GA}^{(\ell)} \rangle_j \approx N\big(\beta(s) - (1-f)\beta\big(\tfrac{s}{1-f}\big)\big)$. This result matches the behavior found [15] for the $U(1)$ case, with the leading order independent of $\ell$ and the equipartition of entanglement entropy restored.

Interestingly, in the $j = 0$ case, using (224) and (226), we find that at the leading order the entanglement entropy at fixed subsystem charge $\ell$ is $\langle S_{GA}^{(\ell)} \rangle_0 \approx fN \log 2$. As this quantity is independent of the subsystem charge, we conclude that in this case, there is equipartition of entanglement for all $\ell$.

## 6.3 Comparison of $G$-local and $K$-local asymptotics

The asymptotics of the average $K$-local entanglement entropy was studied in [21]. The system considered is the same as the one described here in Sec. 5.1, with the additional assumption of vanishing magnetization, $m = 0$, to select symmetry-resolved states. Using a combination of analytical and numerical methods, asymptotic formulas for the thermodynamic limit $N \to \infty$ at fixed $f$ and $s$ were studied. We compare these results for the ones obtained here in Sec. 6.1–6.2 for the $G$-local entanglement entropy of the same symmetry-resolved states.

For maximal spin $j_{\max} = N/2$, i.e., for the case $s = 1$, the Hilbert space takes the form

$$\mathcal{H}_N^{(j_{\max})} = \mathcal{H}_{\mathrm{sym}}^{(j_{\max})} \otimes \mathcal{H}_{GA}^{(\ell_{\max})} \otimes \mathcal{H}_{GB}^{(j_{\max},\ell_{\max})}, \tag{230}$$

i.e., in the decomposition (166) there is a single allowed value of $\ell$, and the dimensions are $\dim \mathcal{H}_{GA}^{(\ell_{\max})} = 1$, $\dim \mathcal{H}_{GB}^{(j_{\max},\ell_{\max})} = 1$, and $\dim \mathcal{H}_{\mathrm{sym}}^{(j_{\max})} = N + 1$. As a result, the average $G$-local entropy necessarily vanishes (219) because any symmetry-resolved state in this sector

has zero entanglement between $G$-local degrees of freedom. On the other hand, the $K$-local entanglement entropy also measures the entanglement in magnetic degrees of freedom $m_A$ at fixed $m = m_A + m_B$ in $\mathcal{H}_{\text{sym}}^{(j_{\max})}$, which are not $G$-local. This $K$-local entanglement results in an average entropy that scales as $\frac{1}{2}\log N$. The comparison of the two different scalings found in [21] and in Sec. 6.2 is shown in the table:

| $j = j_{\max}$ | $N$ | $\sqrt{N}\,\delta_{f,\frac{1}{2}}$ | $\log N$ | $1$ | $\delta_{f,\frac{1}{2}}$ |
|---|---|---|---|---|---|
| $\langle S_{GA}\rangle_{j_{\max}}$ | $0$ | $0$ | $0$ | $0$ | $0$ |
| $\langle S_{KA}\rangle_{j_{\max},m=0}$ | $0$ | $0$ | $\frac{1}{2}$ | $\frac{1}{2}\log\frac{\pi e f(1-f)}{2}$ | $0$ |

(231)

For minimum spin, $j = 0$ (assuming $N$ even), i.e., for the case $s = 0$, the leading order $O(N)$ of the $K$-local and $G$-local average entropies coincide. There is again a difference at order $O(\log N)$ and at order $O(1)$ (first computed in [27] for the $K$-local entropy). A comparison of the contributions found in [21] and in Sec. 6.2 is shown in the table:

| $j = 0$ | $N$ | $\sqrt{N}\,\delta_{f,\frac{1}{2}}$ | $\log N$ | $1$ | $\delta_{f,\frac{1}{2}}$ |
|---|---|---|---|---|---|
| $\langle S_{GA}\rangle_0$ | $f\log 2$ | $0$ | $-\frac{1}{2}$ | $a_{GA}(f)$ | $-\frac{1}{2}$ |
| $\langle S_{KA}\rangle_0$ | $f\log 2$ | $0$ | $0$ | $a_{KA}(f)$ | $-\frac{1}{2}$ |

(232)

where

$$a_{KA}(f) = \begin{cases} \frac{3}{2}f + \frac{3}{2}\log(1-f), & f \leq 1/2, \\ f \longleftrightarrow (1-f), & f > 1/2, \end{cases}$$

(233)

$$a_{GA}(f) = \begin{cases} \frac{3}{2}f + \log(1-f) - \frac{1}{2}\log f + (\frac{1}{2}-f)\log(2) - 1 - \frac{\gamma_E}{2}, & f \leq 1/2, \\ f \longleftrightarrow (1-f), & f > 1/2. \end{cases}$$

(234)

We note the symmetry $f \longleftrightarrow (1-f)$, which is an exact symmetry for $j = 0$.

For spin $j$ of order $O(N)$, i.e., fixed spin density $s = 2j/N$ with $0 < s < 1$, the asymptotics of the $K$-local average entropy studied in [21] and the $G$-local average entropy derived in Sec. 6.1 agree at order $O(N)$ and $O(\sqrt{N})$. A difference again arises at order $O(\log N)$ and $O(1)$, as summarized in the table:

| $0 < s < 1$ | $N$ | $\sqrt{N}\,\delta_{f,\frac{1}{2}}$ | $\log N$ | $1$ | $\delta_{f,\frac{1}{2}}$ |
|---|---|---|---|---|---|
| $\langle S_{GA}\rangle_s$ | $\beta(s)f$ | $-\dfrac{|\beta'(s)|}{\sqrt{2\pi|\beta''(s)|}}$ | $0$ | $b_{GA}(f,s)$ | $c_{GA}(s)$ |
| $\langle S_{KA}\rangle_{s,m=0}$ | $\beta(s)f$ | $-\dfrac{|\beta'(s)|}{\sqrt{2\pi|\beta''(s)|}}$ | $\frac{1}{2}$ | $b_{KA}(f,s)$ | $0$ |

(235)

where $\beta(s)$, $\beta'(s)$, $\beta''(s)$ are given by (205)–(206), and

$$b_{KA}(f,s) = \begin{cases} \frac{f+\log(1-f)}{2} - \frac{1-2f(1-s)}{2s}\log\left(\frac{1+s}{1-s}\right) + \log\frac{2s^{3/2}}{\sqrt{1-s^2}} + \frac{1}{2}\log\frac{\pi e f(1-f)}{2}, & f \leq 1/2, \\ f \longleftrightarrow (1-f), & f > 1/2, \end{cases}$$

(236)

$$b_{GA}(f,s) = \begin{cases} \frac{f+\log(1-f)}{2} - (1-f)\frac{1-s}{2s}\log\left(\frac{1+s}{1-s}\right) + \log\left(\frac{2s}{1+s}\right), & f \leq 1/2, \\ \frac{(1-f)+\log f}{2} + (1-f)\frac{1-s}{2s}\log\left(\frac{1+s}{1-s}\right), & f > 1/2, \end{cases}$$

(237)

$$c_{GA}(f,s) = \frac{1}{2}\frac{1-s}{2s}\log\left(\frac{1+s}{1-s}\right) - \frac{1}{2}\log\left(\frac{2s}{1+s}\right).$$

(238)

We note that the asymmetry of the $G$-local average entropy under subsystem exchange, $f \longleftrightarrow (1-f)$, discussed in Sec. 5.4 and Sec. 6.1 arises only at order $O(1)$, as shown explicitly in (237).

Additionally, we observe that in all the cases considered here, we find that the average $K$-local entropy studied in [21] and the average $G$-local entropy derived here satisfy:

$$\langle S_{KA} \rangle - \langle S_{GA} \rangle = \tfrac{1}{2} \log N + O(1). \tag{239}$$

The difference can be attributed to the entanglement in the magnetic degrees of freedom probed by $K$-local observables, as discussed in (231).

# 7 Discussion

This paper introduces a mathematical framework for symmetry-resolved entanglement with a non-Abelian symmetry group $G$. The framework relies on the notion of subsystem defined operationally in terms of a subalgebra of observables (Sec. 2). In the presence of a non-Abelian symmetry, symmetry-resolved observables that are $G$-invariant determine symmetry-resolved states that can be prepared and measured by these observables, together with a decomposition of the Hilbert space that generalizes the familiar notions of direct sum over Abelian charges and of tensor product over independent subsystems (Sec. 3). The framework is general and does not require a built-in notion of locality or the choice of a Hamiltonian. Once we introduce a notion of locality, a distinction between $K$-local and $G$-local observables arises: kinematical observables in a many-body system are naturally local but, in general, are not invariant under transformations of the group $G$. On the other hand, invariant observables are generally non-local. $G$-local observables are both local and $G$-invariant. In various physical settings, they are the only accessible observables defining a symmetry-resolved subsystem. Symmetry-resolved entanglement entropy is defined here as the entropy $S_{GA}$ of symmetry-resolved states restricted to a symmetry-resolved subsystem (Sec. 4). To illustrate this general framework, we considered the example of a system invariant under the group $G = SU(2)$, computed the exact average entanglement entropy and its variance for random symmetry-resolved states (Sec. 5) and studied its Page curve in the thermodynamic limit (Sec. 6).

In the case of an Abelian symmetry, such as the group $G = U(1)$, the construction presented here reduces to known results [4–8] because the relation (110) significantly simplifies the symmetry-resolved decomposition. In particular, in the analysis of symmetry-resolved entanglement [13–15], the block-diagonal form of the reduced density matrix with $U(1)$ symmetry plays a key role. In fact, the notion of entanglement asymmetry associated with a projected density matrix has been proposed as a measure of symmetry breaking [64]. In the non-Abelian case, the block-diagonal form (77) arises once one identifies the generalization (74) of symmetry-resolved subsystems. The relevant expressions for the $SU(2)$ case are (175) and (166), which are non-trivial for excited states with $j \neq 0$. It would be interesting to use this formulation to extend the analysis of entanglement asymmetry as a probe of $SU(2)$ symmetry breaking [64].

The Page curve was first introduced in [1] as a measure of the typical entanglement entropy of a random pure state without any constraints. The distribution of the typical entropy in the presence of an additive constraint (corresponding to an Abelian symmetry) was introduced in [2], where an exact formula for the average and the variance is given. In this paper, we derived exact formulas for the average and variance of the distribution of the entanglement entropy of random pure states with non-Abelian symmetry group $G$, (43)–(44)–(78). When the results of [2] for an Abelian symmetry are applied to the thermodynamic limit of a many-body system, the average over the distribution reproduces a volume law $V \sim N$ for

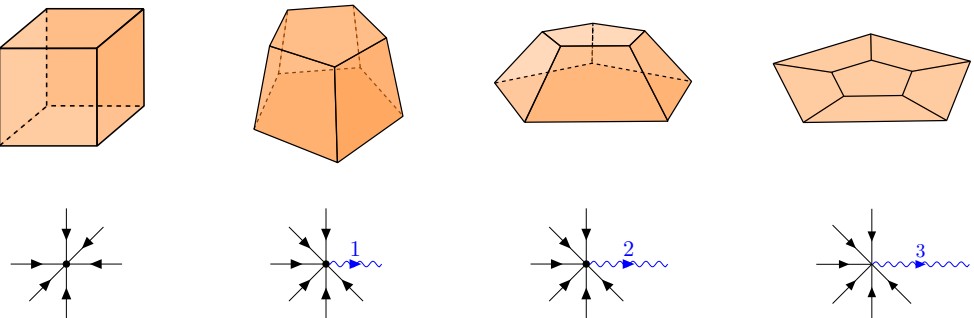

Figure 5: Quantum polyhedra provide a concrete example of $SU(2)$ symmetry-resolved states that can be probed only using $G$-local observables that measure their intrinsic geometry. Each spin corresponds to a quantum plane of fixed area, and the $SU(2)$ invariance in the coupling of angular momenta corresponds to the closure of the faces of the polyhedron [106, 107].

the entanglement entropy [4–8]. Moreover, a $\sqrt{V}$ correction arises at half-system size because of the Abelian constraint. This correction was first identified analytically and observed numerically in [5], then explained in terms of energy conservation in [109] and derived from number conservation in [4]. A global non-Abelian symmetry, such as $SU(2)$, also results in a leading-order volume-law entanglement entropy, together with a square-root-volume correction at half-system size, as first found in [21] and derived in Sec. 6 from the asymptotics of the exact formulas for the typical entropy (See (235) for a comparison). The coefficient (205) of the volume-law scaling depends on the spin density $j/V$, which can be generalized to the densities $q_k/V$ for the non-Abelian charges, i.e., the rank($G$) Casimir operators of a general compact semisimple Lie group $G$. Moreover, at order $O(1)$ in the thermodynamic limit, an asymmetry under subsystem exchange arises, $f \leftrightarrow\!\!\!/ (1-f)$, where $f = V_A/V$ is the subsystem-volume fraction. This is a new feature of non-Abelian symmetry-resolved entanglement that we discuss in Sec. 4.2 for a general group $G$, and illustrate in (196) for a small system and in (237) for a many-body system in the thermodynamic limit. While for a global symmetry, this phenomenon is subleading in the thermodynamic limit, it would be interesting to investigate its consequences in the case of a local symmetry and its impact on the estimate of the Page time in black hole evaporation [24–26, 110–112].

The results of this paper are general and do not depend on the choice of a specific Hamiltonian. They depend only on the structure of the Hilbert space and on the representation of the symmetry group $G$. This group can be understood as a symmetry of the dynamics, such as the Hamiltonian (2) discussed in the introduction as a motivation. Entanglement in eigenstates of a specific quantum-chaotic Hamiltonian with Abelian symmetry group $G = U(1)$ is studied in [9, 10] for the spin-1/2 $XXZ$ chain and in [11, 12] for the $XXZ$ model and the mixed-field Ising model with a constrained energy window. Remarkably, they show that the distribution of the entanglement entropy of mid-spectrum energy eigenstates, computed numerically, agrees with the analytical distribution found in [2]. We conjecture that the distribution $P(S_{GA})$ with average and variance (43)–(44)–(78) found in this paper (See Fig. 3(a)) matches the distribution of the entanglement entropy of energy eigenstates of quantum-chaotic Hamiltonians with non-Abelian symmetry group $G$, restricted to $G$-local subsystems. It would be interesting to test this conjecture numerically, extending the state-of-the-art exact diagonalization of [9–12] to quantum-chaotic Hamiltonians with a non-Abelian symmetry, such as the random Heisenberg model (2) and the Heisenberg model on a lattice with local interactions [18], as done in [20, 21] for $SU(2)$. This conjecture is of direct relevance to recent developments in non-

Abelian eigenstate thermalization [19–21], thermodynamics with noncommuting conserved charges [27, 89–91], and quantum many-body scars [95, 96]. It would also be interesting to extend our framework to the analysis of quantum systems near criticality, random systems where the entanglement entropy behaves effectively as in critical systems [113], the case of WZW models and $SU(2)_k$ symmetry [14, 85–87] and to the case of Virasoro symmetry in a conformal field theory [88], going beyond the special case of the vacuum state.

Another new phenomenon that arises in the non-Abelian case is the distinction (88) between $K$-local and $G$-local subsystems. The notion of $G$-local observables plays an effective role in systems with a gapped Hamiltonian and selection rules that restrict the accessible observables to multiplets in the energy spectrum (See Fig. 1). $G$-local observables also play a fundamental role in systems with an intrinsic symmetry that constrains the accessible observables [47–52], as it is the case for quantum reference frames where one adopts an intrinsic perspective [114–119]. Another example where $G$-local observables play a central role is the case of the quantum geometry of a polyhedron [106, 107]. The $G$-local observables $\vec{S}_a \cdot \vec{S}_b$ described in Fig. 1 measure the angle between the faces of the quantum polyhedron, shown in Fig. 5 for different sectors of spin $j$.

It would be interesting to derive also the exact formulas for the average and the variance of the probability distribution $P(S_{KA})$ for $K$-local observables (See Fig. 3). In [21], the asymptotics of the average was computed for a spin system using a combination of analytical and numerical techniques, and in (239) we observed that the averages of $S_{KA}$ and $S_{GA}$ differ at order log-volume in the thermodynamic limit.

The framework introduced in this paper applies both to groups $G$ that act globally on the system—a rigid symmetry—and to groups $G^{\times N}$ that act locally at the $N$ nodes of a lattice, i.e., a lattice gauge symmetry [41–43]. In this case, we expect that the new features of non-Abelian symmetry-resolved entanglement play a central role, and it would be interesting to explore their effect on the Page curve in lattice gauge theory [120] and spin-network states in loop quantum gravity [44–46, 121–123].

In this paper, we assumed a finite-dimensional Hilbert space, but the results presented apply directly to each symmetry-resolved sector that is a finite-dimensional subspace of a Hilbert space that can be infinite-dimensional. It would be interesting to extend the analysis presented here to quantum field theory [35–39] where, in a finite volume and at fixed energy, the microcanonical sectors of the Hilbert space have finite dimension and the Page curve has been argued to reproduce black-body thermodynamics [2]. Finally, the Page curve was initially introduced as a tool to identify the non-perturbative time scales of black hole evaporation [25, 26]. It would be interesting to apply the methods introduced in this paper to compute the Page curve for the entanglement entropy of the subalgebra of gravitational news [124], symmetry-resolved with respect to the asymptotic symmetries of a black hole at fixed mass and spin.

# Acknowledgments

We thank Lucas Hackl and Marcos Rigol for many useful discussions.

**Funding information** EB acknowledges support from the National Science Foundation, Grant No. PHY-2207851. This work was made possible through the support of the ID# 62312 grant from the John Templeton Foundation, as part of the project "The Quantum Information Structure of Spacetime" (QISS). The opinions expressed in this work are those of the authors and do not necessarily reflect the views of the John Templeton Foundation. PD thanks the Cross-Consortium visiting program of the QISS Project for supporting his stay at the IGC, where

the first part of this work was done. EB also thanks the Cross-Consortium visiting program of the QISS Project for supporting his stay at the IQOQI. RK thanks the Blaumann Foundation for support to participate in the QISS 2023 School where preliminary results of this project were first presented.

## A  Dimension of $SU(2)$ intertwiner spaces

In this section we give a derivation of the dimension of the Hilbert spaces $\mathcal{H}_{GA}^{(\ell)}$, $\mathcal{H}_{GB}^{(j,\ell)}$, and $\mathcal{H}_G^{(j)}$ based on the calculation of the dimension of the $SU(2)$ invariant spaces

$$\nu(j_1,\dots,j_L) = \dim \mathrm{Inv}_G(\mathcal{H}^{(j_1)} \otimes \cdots \otimes \mathcal{H}^{(j_L)}). \tag{A.1}$$

These invariant spaces, also known as intertwiners, are well studied in loop quantum gravity [44,45], where they are the fundamental building blocks for describing quantum geometries [106,107]. Using techniques typical of intertwiner calculations [23], we write the projector to intertwiner space,

$$P : \mathcal{H}^{(j_1)} \otimes \cdots \otimes \mathcal{H}^{(j_L)} \longrightarrow \mathrm{Inv}_G(\mathcal{H}^{(j_1)} \otimes \cdots \otimes \mathcal{H}^{(j_L)}), \tag{A.2}$$

via group averaging

$$P = \int_{SU(2)} D^{(j_1)}(g) \otimes \cdots \otimes D^{(j_L)}(g) \, dg, \tag{A.3}$$

where $D^{(j)}(g)^m{}_{m'}$ are Wigner matrices for the representation with spin $j$ and $dg$ is the Haar measure for the group $SU(2)$. The dimension of the $SU(2)$–invariant space can then be computed as the trace of the identity in this space or, equivalently, the trace of the projector $\mathrm{Tr}P$, i.e., $\nu$ is given by the integral

$$\nu(j_1,\dots,j_L) = \int_{SU(2)} \chi^{(j_1)}(g) \cdots \chi^{(j_L)}(g) \, dg, \tag{A.4}$$

where $\chi^{(j)}(g) = \mathrm{Tr}D^{(j)}(g)$ is the character of the representation with spin $j$. We compute the integral using the class-angle parametrization $g = h\,e^{i\theta\sigma_z}h^{-1}$ of a group element, which results in the character $\chi^{(j)}(g) = \frac{\sin(2j+1)\theta}{\sin\theta}$ expressed as a function of the class angle $\theta$. The Haar measure for class functions $\psi(g) = f(\theta)$ reduces to $\int dg = \frac{2}{\pi}\int_0^{2\pi}(\sin\theta)^2 \, d\theta$.

This paper uses the dimension $\nu$ with $N$ spin-$1/2$ and one spin-$j$. After a few manipulations with elementary trigonometric identities, we find

$$\nu(\tfrac{1}{2},\dots,\tfrac{1}{2},j) = \frac{2^N}{\pi} \int_0^{2\pi} (\sin\theta)(\cos\theta)^N (\sin(2j+1)\theta) \, d\theta. \tag{A.5}$$

Using the residue theorem, we evaluate the integral (A.5) as a contour integral on the unit circle of the complex plane. The only contribution to the integral comes from the pole in the origin

$$\nu(\tfrac{1}{2},\dots,\tfrac{1}{2},j) = -\tfrac{1}{2}\mathrm{Res}_{z=0}\left(\tfrac{1}{z}\left(\tfrac{1}{z}-z\right)\left(\tfrac{1}{z^{2j+1}}-z^{2j+1}\right)\left(\tfrac{1}{z}+z\right)^N\right). \tag{A.6}$$

We expand the binomial $\left(\tfrac{1}{z}+z\right)^N = \sum_{k=0}^N \binom{N}{k}z^{2k-N}$ and read the residue from the coefficient of $\tfrac{1}{z}$. If $N + 2j$ is odd, the integral vanishes. If $N + 2j$ is even, we find

$$\nu(\tfrac{1}{2},\dots,\tfrac{1}{2},j) = \frac{2j+1}{\frac{N}{2}+j+1}\binom{N}{\frac{N}{2}+j}. \tag{A.7}$$

This is the expression also found, for instance, in [20,21] and derived via combinatorial methods.

Using the intertwiner methods discussed above, we can now compute the dimension $\nu$ with $N - N_A$ spin-1/2, one spin-$j$, and one spin-$\ell$ which appears in the formula for the non-Abelian symmetry-resolved entanglement entropy. We find

$$\nu(\tfrac{1}{2}, \dots, \tfrac{1}{2}, j, \ell) = \frac{2^{N-N_A+1}}{\pi} \int_0^{2\pi} (\cos\theta)^{N-N_A} \big(\sin(2j+1)\theta\big)\big(\sin(2\ell+1)\theta\big)\, \mathrm{d}\theta \tag{A.8}$$

$$= \binom{N-N_A}{\frac{N-N_A}{2}+j-\ell} - \frac{\frac{N-N_A}{2}-j-\ell}{\frac{N-N_A}{2}+j+\ell+1}\binom{N-N_A}{\frac{N-N_A}{2}+j+\ell}. \tag{A.9}$$

We note that this formula reduces to the expression (A.7) for $\ell = 0$ or $j = 0$.

To summarize, we have

$$D_j = \dim \mathcal{H}_G^{(j)} = \nu(\tfrac{1}{2}, \dots, \tfrac{1}{2}, j), \tag{A.10}$$

$$d_\ell = \dim \mathcal{H}_{GA}^{(\ell)} = \nu(\tfrac{1}{2}, \dots, \tfrac{1}{2}, \ell), \tag{A.11}$$

$$b_{j\ell} = \dim \mathcal{H}_{GB}^{(j,\ell)} = \nu(\tfrac{1}{2}, \dots, \tfrac{1}{2}, j, \ell). \tag{A.12}$$

# B  Laplace approximation and discontinuities

In this section, we review the asymptotic expansion of integrals using the Laplace method [125] and extend this method to the presence of discontinuities. We derive an asymptotic expansion for $N \gg 1$ of integrals of the form

$$I = \int_K h(x)\, \mathrm{e}^{Ng(x)}\, \mathrm{d}x, \tag{B.1}$$

where $K \subset \mathbb{R}$ and $h(x)$ and $g(x)$ are two real-valued functions defined on $K$ such that the integral is well defined for large enough $N$. We first assume that the functions $h$ and $g$ are smooth on $K$. Then, we present the formula for the case with $h$ continuous but with a discontinuous first derivative at a point. We assume that the function $g$ has a global maximum at $x_0$ in $K$ where the gradient vanishes, $g'(x_0) = 0$, and the function also vanishes at the maximum $g(x_0) = 0$. The integral is approximated asymptotically by

$$I = h(x_0)\sqrt{-\frac{2\pi}{Ng''(x_0)}}\left(1 + \frac{C_1}{N} + o\left(\frac{1}{N}\right)\right), \tag{B.2}$$

where

$$C_1 = -\frac{1}{2}\frac{h''(x_0)}{h(x_0)g''(x_0)} + \frac{1}{8}\frac{g''''(x_0)h(x_0)}{h(x_0)g''(x_0)^2} + \frac{1}{2}\frac{g'''(x_0)h'(x_0)}{h(x_0)g''(x_0)^2} - \frac{5}{24}\frac{g'''(x_0)^2 h(x_0)}{h(x_0)g''(x_0)^3}. \tag{B.3}$$

The approximation (B.2) relies on three main observations.

1. Only the immediate neighborhood of $x_0$ contributes to the integral.

2. We expand the function $g(x)$ around $x_0$, and we approximate the exponential part of the integrand with a narrow Gaussian $e^{-\frac{N}{2}g''(x_0)(x-x_0)^2}$. This confirms the first assumption.

3. We expand the remaining part of the integrand around $x_0$ and extend the integral to the whole real line.

The integrals of terms of the expansion with the Gaussian ($x_0$ is a maximum, so $g''(x_0) < 0$) are given by

$$M^{(p)} = \int_{-\infty}^{\infty} (x-x_0)^p\, e^{N\frac{g''(x_0)}{2}(x-x_0)^2} dx = \begin{cases} 0, & p \text{ odd}, \\ \sqrt{2\pi}(p-1)!!\left(-\frac{1}{Ng''(x_0)}\right)^{\frac{p+1}{2}}, & p \text{ even}. \end{cases} \tag{B.4}$$

If we are interested in the leading order or (B.2), expanding the integrand up to order $O(1)$ and combining with $M_0$ is sufficient

$$h(x_0)M^{(0)} = h(x_0)\sqrt{-\frac{2\pi}{Ng''(x_0)}}. \tag{B.5}$$

However, to compute the next-to-leading order (B.3), we have to expand the integrand up to order $O(x-x_0)^6$. The relevant terms are

$$h''(x_0)M^{(2)} = -\frac{1}{N}\frac{1}{2}\frac{h''(x_0)}{g''(x_0)}\sqrt{-\frac{2\pi}{Ng''(x_0)}},$$

$$N\left(\frac{1}{24}g''''(x_0)h(x_0) + \frac{1}{6}g'''(x_0)h'(x_0)\right)M^{(4)} = \frac{1}{N}\left(\frac{1}{8}\frac{g''''(x_0)}{g''(x_0)^2} + \frac{1}{2}\frac{g'''(x_0)h'(x_0)}{h(x_0)g''(x_0)^2}\right)\sqrt{-\frac{2\pi}{Ng''(x_0)}},$$

$$N^2\frac{1}{72}g'''(x_0)^2h(x_0)M^{(6)} = -\frac{1}{N}\frac{5}{24}\frac{g'''(x_0)^2}{g''(x_0)^3}\sqrt{-\frac{2\pi}{Ng''(x_0)}}.$$

Collecting them together, we obtain (B.3).

In the derivations in Sec. 6, we need to compute integrals of the form (B.1) with a function $h$ that is not continuous in $x_0$. For simplicity, we will assume that $g(x)$ is still smooth in $x_0$. We follow the same strategy, expanding the integrand around $x_0$ and separating the integrals in $x < x_0$ and $x > x_0$. The split Gaussian integrals are such that

$$\int_{x_0}^{\infty} (x-x_0)^p\, e^{N\frac{g''(x_0)}{2}(x-x_0)^2} dx = (-1)^p \int_{-\infty}^{x_0} (x-x_0)^p\, e^{N\frac{g''(x_0)}{2}(x-x_0)^2} dx, \tag{B.6}$$

and

$$\tilde{M}^{(p)} = \int_{x_0}^{\infty} (x-x_0)^p\, e^{N\frac{g''(x_0)}{2}(x-x_0)^2} dx = \begin{cases} \frac{1}{2}\sqrt{2\pi}(\frac{p-1}{2})!\left(-\frac{1}{Ng''(x_0)}\right)^{\frac{p+1}{2}}, & p \text{ odd}, \\ \frac{1}{2}\sqrt{2\pi}(p-1)!!\left(-\frac{1}{Ng''(x_0)}\right)^{\frac{p+1}{2}}, & p \text{ even}. \end{cases} \tag{B.7}$$

The leading order of the Laplace approximation is similar to the one found earlier,

$$\left(h(x_0^-) + h(x_0^+)\right)\tilde{M}^{(0)} = \frac{h(x_0^-) + h(x_0^+)}{2}\sqrt{-\frac{2\pi}{Ng''(x_0)}}. \tag{B.8}$$

Here, we denote with $h(x_0^\pm)$ the right/left limit of the function $h$ in $x_0$. If $h$ is continuous in $x_0$, the formula (B.8) reduces to (B.5). The next-to-leading order term is

$$\left(h'(x_0^-) - h'(x_0^+)\right)\tilde{M}^{(1)} = \frac{h'(x_0^-) - h'(x_0^+)}{2}\sqrt{2\pi}\frac{-1}{Ng''(x_0)}, \tag{B.9}$$

which is, in general, non-vanishing if the first derivative of $h$ is discontinuous in $x_0$. The calculation for the next-to-next-leading order term is similar. To summarize, if the function $h$ is discontinuous at the maximum $x_0$ of $g$, the Laplace approximation for the integral (B.1) is

$$I = \frac{h(x_0^-) + h(x_0^+)}{2}\sqrt{-\frac{2\pi}{Ng''(x_0)}}\left(1 + \frac{\tilde{C}_{1/2}}{\sqrt{N}} + \frac{\tilde{C}_1}{N} + o\left(\frac{1}{N}\right)\right), \tag{B.10}$$

where

$$\tilde{C}_{1/2} = \frac{h'(x_0^-) - h'(x_0^+)}{h(x_0^-) + h(x_0^+)} \sqrt{-\frac{2}{N\pi g''(x_0)}}, \tag{B.11}$$

and

$$\tilde{C}_1 = \frac{h(x_0^-)C_1(x_0^-) + h(x_0^+)C_1(x_0^+)}{h(x_0^-) + h(x_0^+)}, \tag{B.12}$$

where we denote by $C_1(x_0^\pm)$ the expression (B.3) computed (as a limit) in $x_0^\pm$. Therefore, a discontinuity results in a $O(1/\sqrt{N})$ correction in the asymptotic expansion.

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
