# Peer review of "Non-abelian symmetry-resolved entanglement entropy"

_SciPost Physics, doi:SciPost Phys. 17, 127 (2024)_

## Round 1 · Referee Report · Anonymous (Referee 1) · 2024-5-31

Strengths

  1. The paper provides a novel description of entanglement measures in systems possessing an internal non-abelian symmetry.
  2. The paper generalises known entanglement measures and the notion of locality, to non-abelian symmetries.
  3. The paper provides relevant analytical and numerical examples.
  4. The paper introduces a more mathematically rigorous treatment of the problem.
  5. The paper is very well written.

Weaknesses

  1. The applicability of the results to known models could be made clearer.

Report

I read this paper with great interest and I think it is a high quality paper. Clearly a lot of effort has gone into the writing, editing, the quality of figures and presentation overall, which I really appreciate. I think that the paper meets all the requirements for publication in SciPost.

The paper provides a new viewpoint and mathematical approach to defining symmetry resolved entanglement measures in systems with non-abelian symmetries, leaving scope for further generalisations and applications to various models. It proposes a novel treatment and definition of useful entanglement measures in this context, by placing particular attention on the choice of entanglement state and on the definition of locality of the entanglement region. These are all very interesting and deep concepts, in the context of entanglement measures.

The paper is is well written, provides a good literature review and comprehensive set of citations, contains a lot mathematical details as well as explicit examples for specific models. Finally, the Appendix provides further details on the more intricate computations, such those related to the derivation of asymptotic properties.

All in all, the paper could be published as is. I have some questions/suggestions that I present below, but I would consider these minor changes that the authors may want to think about.

Requested changes

As I said in the report, I don't really have a request for any major changes. However, there is something that I noticed which perhaps could be improved and would help make the paper more appealing to people like me, which are less mathematically inclined and more focused on analytical computations.

It strikes me that because most notions in the paper are introduced from the mathematical view point of symmetry groups and symmetry algebras, there is very little discussion of how the physical nature of the system can affect the results. What I mean mainly is, for instance, whether it makes any difference that the system is critical or not. Given that so many studies of this quantity have been done for 1+1D critical and non-critical (mainly integrable) systems I think that it would be useful to highlight how the nature of the correlation length (infinite or finite) affects the results. Where/how does it enter?

For example, the first model that is mentioned is the Ising chain with random couplings. Such a system is not critical in the usual sense but it is known from studies of more standard entanglement measures, that many such measures (on average) scale as in critical systems, with the central charge replaced by an "effective" central charge. See e.g. the review https://arxiv.org/abs/0908.1986. So this chain is a rather special example, even before considering the role of any symmetries.

Recommendation

Publish (surpasses expectations and criteria for this Journal; among top 10%)

  • validity: top
  • significance: high
  • originality: top
  • clarity: top
  • formatting: perfect
  • grammar: excellent

Author:  Eugenio Bianchi  on 2024-10-03  [id 4829]

(in reply to Report 1 on 2024-05-31)
Category:
answer to question

We thank the referee for their comments and their overall positive assessment of the manuscript.

To help identifying the changes, we attach a pdf of the manuscript where we highlighted in red the modifications.

In particular, we added a reference to 0908.1986 [Refael:2009J] as it might result in interesting applications. We included also a remark on extending our analysis to random systems where the entanglement entropy behaves effectively as in critical systems (Sec.7, page 43).

Attachment:

Non-Abelian-Symmetry-Resolved-Entanglement-scipost-v2-red.pdf

---

## Round 1 · Referee Report · Anonymous (Referee 2) · 2024-6-22

Strengths

1-Formal and precise analysis of the structure of the entanglement in the presence of non-abelian symmetries.

2-Well-written introduction with a summary of the main notions studied in the manuscript.

Weaknesses

Lack of comparison with the existing literature about symmetry resolution and comments on the physical aspects of the problem.

Report

In this manuscript, the authors study in great detail the structure of the Hilbert space, states, subsystems, observables and entanglement in the presence of non-abelian symmetries. They point out an interesting factorization between the algebra of $K$-local observables, that in general are not invariant under transformation of the symmetry group, and $G$-local observables, which are both local and invariant. The $G$-local observables in a subsystem $A$ are associated with a density matrix $\rho_{GA}$ and the authors define the symmetry-resolved entanglement (SRE) as the entanglement associated with this density matrix.

I am slightly confused between the definition of the SRE used in this paper and the one introduced in Ref. [13]. Is it correct to say that, in the framework of [13], the SRE should be associated to $\rho_{GA}^{j_A}$, rather than $\rho_{GA}$? In other words, Eq. 9 shows the decomposition of the entanglement of $\rho_{GA}$ into 2 contributions, known as configurational and fluctuation entanglement, and the SRE is given by $-\mathrm{Tr}(\rho_{GA}^{j_A}\log \rho_{GA}^{j_A})$. The symmetry resolution is usually done with respect to the charge of the subsystem ($j_A$ in this case), not of the total system ($j$).

I find the paper very interesting for people working on entanglement and symmetries, however, in my opinion, even though they authors report a comprehensive set of citations, there are not comparisons of their results with the ones existing in the literature. For instance, how do their results compare with the symmetry-resolution studied in the presence of a $U(1)$ Haar-random ensemble in Ref. [5] (there is only a short sentence in the conclusion)? And with the results for the non-abelian symmetry resolution in critical systems ([13,83,84, JHEP 2023, 216 (2023)])? Another recurrent question in the study of the symmetry resolution is the equiparition of the entanglement (i.e. Ref. [14]). It would be helpful if the authors could comment it, especially in the order of limits they consider here (i.e. with a fixed subsystem fraction). In my opinion, these discussions related more to the physical aspect of the problem could make the results even more interesting for readers who are not familiar with the mathematical language used in this paper.

Therefore, I would recommend the paper for publication only after the following changes.

Requested changes

1-As I have mentioned in the report, I understand that the symmetry-resolution of the entanglement in the presence of a non-abelian symmetry can be defined in terms of the irreducible represesentation of the symmetry group, but the authors should comment the connection between the entanglement of $\rho_{GA}^{j_A}$ and $\rho_{GA}$. Given that the nomenclature 'symmetry-resolution' also enters in the title of the manuscript, it would be appropriate avoiding conflicts with the existing literature. 2-More in general, it would be beneficial commenting more their findings in light of the results about the symmetry resolution of the entanglement. 3- An approach to symmetry resolution in the context of operator algebras has been considered in 2305.02343, where the authors study the resolution of the modular flow of operators belonging to algebras of invariant observables. Would it be correct to claim that the authors of the present manuscript consider an algebraic setup analogous to the one in 2305.02343, focusing on the 'resolution' of a different quantity, namely the entanglement entropy?

Recommendation

Ask for minor revision

  • validity: good
  • significance: high
  • originality: good
  • clarity: good
  • formatting: good
  • grammar: excellent

Author:  Eugenio Bianchi  on 2024-10-03  [id 4830]

(in reply to Report 2 on 2024-06-22)
Category:
answer to question

We thank the referee for their careful reading of the paper. We believe that, thanks to their several comments and suggestions, we have improved the quality and clarity of the manuscript by connecting it more directly to the existing literature about symmetry resolution and adding comments on the physical aspects of the problem.

To help identifying the changes, we attach a pdf of the manuscript where we highlighted in red the modifications.

Requested Changes:

1- As I have mentioned in the report, I understand that the symmetry-resolution of the entanglement in the presence of a non-abelian symmetry can be defined in terms of the irreducible representation of the symmetry group, but the authors should comment the connection between the entanglement of $\rho^{(j)}_{GA}$ and $\rho_{GA}$. Given that the nomenclature 'symmetry-resolution' also enters in the title of the manuscript, it would be appropriate avoiding conflicts with the existing literature.

Section 1, page 6 We have added a paragraph along with a new table, describing the different types of entropies referred to in the paper.

Section 4.2, page 22 We significantly extended section 4.2 on the general case of G-local entanglement entropy $S_{GA}$. We added a discussion with relevant formulae on the difference between the entanglement entropies of $\rho_{GA}$ and $\rho^{(j)}_{GA}$. We provide also a new formula for the average sub-system symmetry-resolved, configurational and number entropies in the case of a general non-abelian symmetry group.

2a- More in general, it would be beneficial commenting more their findings in light of the results about the symmetry resolution of the entanglement. [...] For instance, how do their results compare with the symmetry-resolution studied in the presence of a U(1) Haar-random ensemble in Ref. [5] (there is only a short sentence in the conclusion)?

Section 4.3, page 24 We significantly extended section 4.3 on the special case of U(1) symmetry-resolved entanglement, by connecting our formalism to the one already present in the literature on Haar-random ensemble (Ref.[5]) and about entanglement equipartition (Ref.[14]). In particular, we derive the relevant formulae for the entanglement entropies of $\rho_{GA}$ and $\rho^{(j)}_{GA}$ and the configurational and number entropies. We find that our result exactly matches the ones obtained in Ref[5].

2b- Another recurrent question in the study of the symmetry resolution is the equiparition of the entanglement (i.e. Ref. [14]). It would be helpful if the authors could comment it, especially in the order of limits they consider here (i.e. with a fixed subsystem fraction).

Section 6.2, page 40 We have included a new discussion on (lack of) equipartition of entanglement in the thermodynamic limit of $SU(2)$ symmetry-resolved entanglement. This section now extends the analysis of the order of limits (i.e. with a fixed subsystem fraction) from the $U(1)$ case to the non-abelian $SU(2)$ case.

2c- [...] And with the results for the non-abelian symmetry resolution in critical systems ([13,83,84, JHEP 2023, 216 (2023)])?

Section 7, page 43 We added a reference to [Kusuki:2023bsp] and we included a remark in the discussion about extending our framework to the case of WZW models.

3- An approach to symmetry resolution in the context of operator algebras has been considered in 2305.02343, where the authors study the resolution of the modular flow of operators belonging to algebras of invariant observables. Would it be correct to claim that the authors of the present manuscript consider an algebraic setup analogous to the one in 2305.02343, focusing on the 'resolution' of a different quantity, namely the entanglement entropy?

We added a reference to 2305.02343 [DiGiulio:2023nvz] as it uses a similar language, even though it considers a different setup. The algebraic setup considered there focuses on vacuum correlations and the local algebra of operators that preserve a U(1)-symmetric vacuum. On the other hand, our setup focuses on the operational definition of subsystems in terms of a subalgebra of observables compatible with the (non-abelian) symmetries of the system, without any reference to the vacuum state.

Attachment:

Non-Abelian-Symmetry-Resolved-Entanglement-scipost-v2-red_lDfrqPl.pdf

---

## Round 2 · Referee Report · Anonymous (Referee 2) · 2024-10-6

Report

I thank the authors for implementing the requested changes. I think that the further explanations and the comparisons with the existing literature have significantly improved the clarity of the paper and I recommend it for publication.

Recommendation

Publish (easily meets expectations and criteria for this Journal; among top 50%)

---

## Round 2 · Referee Report · Anonymous (Referee 1) · 2024-10-6

Strengths

Same as in my first report

Weaknesses

The main weakness for me was that the mathematical nature of the work obscured the physical applications. This has been improved considerably in the 2nd version.

Report

The authors have submitted a revised version of their paper which answers my (minor) comments and those of the other referee. We both had rather similar comment in fact and the authors have made a considerable effort to comment on those, in particular to explain more clearly what they compute and how it might apply to specific types of theories. I recommend that the paper is now published.

Requested changes

None.

Recommendation

Publish (easily meets expectations and criteria for this Journal; among top 50%)

---

## Round 2 · Author Response

We thank the referees for their their careful reading of the manuscript and the feedback that helped us improve the presentation. In the new version, we have carefully addressed each of the concerns and comments raised by the referees by expanding significantly a subsection, adding a new table, adding references, and including clarifying remarks that improve the clarity of the presentation. In the replies, we provide a detailed response to each of the referees's comments.

---

## Round 2 · List of Changes

List of changes: - Added remarks; - Added table; - Typos fixed; - References added. To help identifying the changes we highlighted them in red in the version attached to the referee responses.

---

## Editorial Decision

published